# FuXi-Ocean: A Global Ocean Forecasting System with Sub-Daily Resolution

**Qiusheng Huang**[1,2†]**, Yuan Niu**[3†]**, Xiaohui Zhong**[1]**, Anboyu Guo**[1]**, Lei Chen**[1]**,**
**Dianjun Zhang**[3]**, Xuefeng Zhang**[3,∗]**Hao Li**[1,2∗]

[1]Artificial Intelligence Innovation and Incubation Institute, Fudan University
[2]Shanghai Innovation Institute
[3]School of Marine Science and Technology,Tianjin University

## Abstract

Accurate, high-resolution ocean forecasting is crucial for maritime operations and environmental monitoring. While traditional numerical models are capable of producing sub-daily, eddy-resolving forecasts, they are computationally intensive and face challenges in maintaining accuracy at fine spatial and temporal scales. In contrast, recent data-driven approaches offer improved computational efficiency and emerging potential, yet typically operate at daily resolution and struggle with sub-daily predictions due to error accumulation over time. We introduce FuXi-Ocean, the first data-driven global ocean forecasting model achieving six-hourly predictions at eddy-resolving $1/12°$ spatial resolution, reaching depths of up to 1500 meters. The model architecture integrates a context-aware feature extraction module with a predictive network employing stacked attention blocks. The core innovation is the Mixture-of-Time (MoT) module, which adaptively integrates predictions from multiple temporal contexts by learning variable-specific reliability , mitigating cumulative errors in sequential forecasting. Through comprehensive experimental evaluation, FuXi-Ocean demonstrates superior skill in predicting key variables, including temperature, salinity, and currents, across multiple depths.

## 1 Introduction

Ocean forecasting systems play a vital role in maritime operations, providing critical information for navigation, search and rescue, fisheries management, and offshore energy production. These applications require increasingly accurate predictions at finer temporal and spatial resolution to capture rapidly evolving oceanic phenomena. For instance, high-resolution forecasts of ocean currents are indispensable for maritime search and rescue and oil spill tracking, where accurate backtracking and source identification are critical [41]. Despite notable advances in numerical modeling and computational capacity, achieving both high temporal resolution and global coverage remains challenging [16, 13].

Operational ocean forecasting traditionally relies on physics-based numerical models that solve the governing equations of ocean dynamics [15]. While based on well-established physical laws, these models are computationally expensive, particularly when resolving mesoscale eddies, which require grid resolutions of $1/12°$ or finer. Even after decades of development, ocean general circulation models (OGCMs) still exhibit substantial uncertainties, due to numerical discretization errors [1] and uncertainties in parameterizing unresolved subgrid processes [34].

---

*Corresponding author.
†These authors contributed equally to this work.

39th Conference on Neural Information Processing Systems (NeurIPS 2025).

Recently, deep learning approaches emerge as promising alternatives to traditional OGCMs. Studies [45, 43, 46, 2, 11] demonstrate that such models can achieve comparable or even superior accuracy at daily temporal resolution compared to operational numerical systems, and their operational efficiency can be improved by a thousand times. However, extending these models to sub-daily resolutions presents unique challenges. Ocean variables exhibit diverse temporal dynamics that vary significantly across depths and spatial regions. Surface variables, such as sea surface temperature, often follow pronounced diurnal cycles similar to atmospheric variables, whereas deeper ocean currents evolve on considerably slower timescales. Unlike atmospheric forecasting, which is dominated by high-frequency variability, ocean prediction must accommodate a broad spectrum of temporal behaviors, from fast-changing surface processes to slowly evolving deep-ocean processes. A key limitation of existing deep learning-based ocean forecasting models lies in their predominant focus on daily forecasts, which restricts their ability to adaptively model variable-specific temporal dynamics at sub-daily intervals. Lacking mechanisms to adaptively process temporal information across multiple timescales, these models often struggle with the complexity of high-frequency sub-daily predictions. Moreover, designing effective sub-daily models requires architectures capable of learning efficiently from limited historical data while avoiding overfitting to spurious correlations.

In this work, we present FuXi-Ocean, the first deep learning-based global ocean forecasting model to achieve six-hour temporal resolution at eddy-resolving 1/12° spatial resolution. Our approach explicitly addresses the challenges of sub-daily ocean prediction through three key innovations: First, we design an autoregressive architecture specifically tailored to capture multi-scale temporal dependencies of different oceanic variables across an unprecedented depth range (0-1500 m). Unlike previous models that treat all variables uniformly, our model adaptively learns temporal context appropriate for each variable and region, effectively distinguishing between fast-evolving surface processes (e.g., diurnal warming) and slowly varying deep-ocean dynamics. Second, we introduce the Mixture-of-Time (MoT) module that adaptively integrates predictions from multiple temporal windows based on their empirical reliability for each variable. This mechanism enables the model to select the most informative temporal context, thereby mitigating the accumulation of forecast errors typically encountered in sequential prediction tasks. Third, we demonstrate remarkable data efficiency, achieving state-of-the-art performance with only 9 years of training datasignificantly less than required by comparable models. This efficiency stems from our architecture's ability to effectively leverage physical constraints and spatial coherence in the learning process. Our key contributions are as follows:

- We propose FuXi-Ocean, the first data-driven global ocean forecasting model to achieve six-hour temporal resolution, 1/12° spatial resolution, and 0-1500 m depth coverage.

- We introduce the Mixture-of-Time (MoT) module, which adaptively integrates variable-specific temporal dependencies to reduce cumulative errors in sequential prediction.

- We validate FuXi-Ocean with reanalysis and observational datasets, demonstrating superior performance over traditional numerical forecasting models at sub-daily intervals.

## 2 Related Work

### 2.1 Numerical Models

Ocean forecasting traditionally relies on OGCMs that solve fundamental physical equations governing ocean dynamics. State-of-the-art operational systems, such as the HYbrid Coordinate Ocean Model (HYCOM) [8, 3], Ocean Physical System (PSY4) [27], Global Ice Ocean Prediction System (GIOPS) [42], Forecast Ocean Assimilation Model (FOAM) [5], BLUElinK OceanMAPS (BLK) [40], Nucleus for European Modelling of the Ocean (NEMO) [19], Modular Ocean Model (MOM) [17], Real-Time Ocean Forecast System (RTOFS) [14] and GLORY12 [22], make significant advances in global ocean prediction capabilities.

Despite their solid theoretical foundations, these models face persistent challenges. The computational cost increases dramatically with resolution, making global simulations at eddy-resolving scales particularly expensive [36]. Additionally, uncertainties in parameterizing unresolved processes, such as vertical mixing and air-sea interactions, introduce systematic model biases [44, 21]. The inherent chaotic nature of ocean systems further complicates forecasting, as small errors in initial conditions can rapidly amplify through nonlinear interactions. These limitations become particularly acute for

high-frequency predictions, where sub-daily forecasts reveal the full spectrum of model deficiencies that might otherwise be obscured in daily averages.

## 2.2 Data-Driven Deep Learning Models

Recent advances in deep learning introduce promising new approaches for ocean forecasting and successfully applied to the global medium-range forecasts [24, 9, 26, 4, 33, 6, 37].At present, deep learning based ocean forecasting technology is developing rapidly and can predict ocean variables such as satellite sea surface temperature, sea surface height, wave height, temperature, salinity, U and V [47, 38], as well as seasonal or interannual ocean phenomena such as Indian Ocean Dipole (IOD) and ElNiño Southern Oscillation (ENSO) [20, 48, 49]. Wang et al. [43] developed "XiHe", a global forecasting model based on the Swin-Transformer architecture, incorporating specialized ocean-land mask mechanisms. Yang et al. [46] introduced "Langya", which mitigates cumulative forecast errors by employing a Time Embedding module, enabling forecasts for up to 7 days without iterative steps. Both models achieve state-of-the-art performance for daily predictions at 1/12° resolution. In addition, Cui et al. [11] and Xiong et al. [45] also experimented with autoregressive forecasting architectures and achieved good performance.

Despite these advances, existing data-driven ocean forecasting approaches face three key limitations that constrain their practical utility. First, temporal resolution remains limited to daily intervals. Ocean parameters evolve on markedly different timescales, with surface temperatures exhibiting pronounced diurnal cycles while deep currents evolving more slowly over extended time periods. Without mechanisms to adaptively learn these multiscale temporal dependencies, current models are fundamentally limited in their ability to capture the full spectrum of ocean dynamics at sub-daily resolutions. Second, vertical coverage in existing models remains severely constrained, typically not extending beyond 700 m depth. The thermocline structure and deep water mass properties below this depth play crucial roles in energy transfer and climate dynamics, yet remain largely unaddressed in current data-driven frameworks. Third, most current approaches rely heavily on atmospheric forcing as input variables, creating additional computational overhead and external data dependencies.

## 3 Method

This section details our methodology for high-frequency ocean forecasting. We first describe our data preparation approach, then present the architecture of FuXi-Ocean, including our novel Mixture-of-Time module for adaptive temporal modeling, and finally outline our training strategy.

### 3.1 Problem Formulation

Ocean forecasting requires processing high-dimensional spatiotemporal data that captures the complex dynamics of global marine systems. FuXi-Ocean addresses the task of predicting future oceanic states from sequential historical observations with high spatial and temporal precision. Formally, given a sequence of historical observations spanning $N$ consecutive time points $\{X^{t-N+1}, X^{t-N+2}, \ldots, X^t\} \in \mathbb{R}^{N \times C \times H \times W}$, we aim to predict the state at the subsequent time step $X_{t+1} \in \mathbb{R}^{C \times H \times W}$. Here, $H = 2160$ and $W = 4320$ represent the spatial dimensions of our global grid at 1/12° resolution, providing eddy-resolving capability essential for capturing mesoscale ocean dynamics.

We focus on five fundamental ocean state variables that collectively characterize the physical state of the global ocean: temperature (T), salinity (S), zonal and meridional components of ocean currents (U and V), and sea surface height (SSH). While SSH is inherently a surface variable, the remaining four variables exhibit substantial vertical variability that necessitates prediction across multiple depth levels. We discretize the water column into 20 strategically selected depth levels from the surface (0m) to the deeper ocean (1500 m): 0, 2, 4, 6, 10, 20, 30, 40, 50, 60, 70, 80, 100, 125, 150, 200, 300, 500, 1000, and 1500 m. This vertical resolution is particularly dense near the surface to accurately capture upper-ocean dynamics that strongly influence air-sea interactions. Consequently, our input and output tensors comprise $C = 81$ channels (4 variables × 20 depth levels + SSH).

The temporal resolution of our data is set at 6-hour intervals, aligning with operational oceanic forecasting standards and balancing computational feasibility with information density. We empirically determine that $N = 4$ consecutive time steps (spanning 24 hours) provides sufficient historical

context to accurately capture diurnal cycles and inertial oscillations while maintaining computational efficiency. To preserve physical coherence in predictions, we apply a land-sea mask to all variables, ensuring the model only processes and predicts values for ocean grid points.

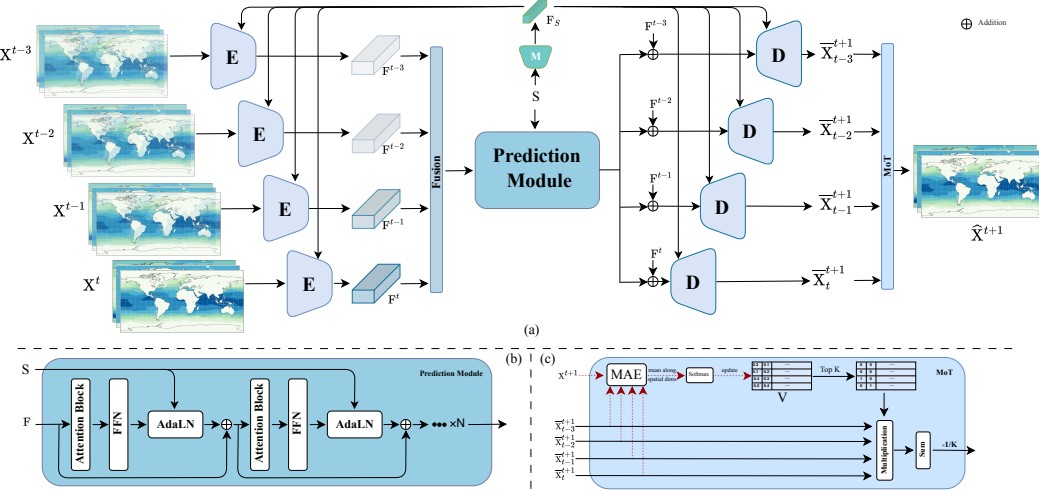

Figure 1: **Architecture of our ocean forecasting framework.** Our model processes sequential ocean states $X^{t-3}$ through $X^t$ to predict $\widehat{X}^{t+1}$. (a) The main pipeline consists of a shared encoder $\mathbf{E}$ that transforms input states into latent representations, modulated by spatiotemporal features $F_S$ from network $\mathbf{M}$. The fused representations feed into the prediction module, whose outputs are processed by decoders $\mathbf{D}$ with skip connections. (b) The prediction module employs stacked attention blocks with adaptive layer normalization (AdaLN) and feed-forward networks (FFN), capturing complex temporal dynamics. (c) The Mixture-of-Time (MoT) module performs channel-wise selection across the four decoder outputs from different temporal skip connections. For each channel, MoT identifies the top-K temporal dependencies using matrix V (derived from spatially-averaged MAE metrics and softmax) and computes the optimal weighted average to synthesize the final prediction $\widehat{X}^{t+1}$.

## 3.2 Model Architecture

FuXi-Ocean employs an autoregressive architecture designed to capture the multiscale temporal dynamics of oceanic variables. The model comprises three primary components: a feature extraction module that encodes multi-temporal inputs, a prediction module that captures temporal evolution patterns, and a feature remapping module that reconstructs the target variables. Figure 1 provides an overview of the model architecture.

### 3.2.1 Feature Extraction

Ocean forecasting requires robust feature representations that capture both spatial dependencies and temporal correlations. We design a feature extraction pipeline that combines context-aware encoding with efficient feature fusion, shown in Fig. 1(a).

Our design employs a shared encoder $\mathbf{E}$ for patch embedding. This encoder uses convolutional layers with matched kernel size and stride for spatial downsampling, followed by layer normalization. This approach balances computational efficiency with representation power when processing global ocean data. To leverage spatiotemporal context, we implement a prior information network $\mathbf{M}$ that processes:

$$F_S = \mathbf{M}(S) \tag{1}$$

where S contains temporal information (diurnal, seasonal patterns) and spatial data (coordinates, bathymetry). The network $\mathbf{M}$ uses sinusoidal positional encodings for temporal components and learnable embeddings for spatial coordinates. We then modulate the encoder weights using these contextual features through:

$$F^t = \mathbf{E}(X^t, F_S) = \mathbf{Norm}(\mathbf{Conv}(X^t, \mathbf{W} \odot F_S)) \tag{2}$$

where W represents the learnable convolution parameters and $\odot$ denotes element-wise multiplication. This modulation enhances the encoder's sensitivity to region-specific patterns, such as boundary currents or seasonal mixed layer dynamics. The encoder processes each input time step $t - i$ ($i \in \{0, 1, 2, 3\}$) to produce feature tensors $\mathbf{F}^{t-i} \in \mathbb{R}^{C' \times H' \times W'}$. Here, $C'$ is the feature dimension while $H'$ and $W'$ represent the downsampled spatial dimensions.

Our feature fusion module concatenates these tensors along the channel dimension, preserving their temporal characteristics. We then apply $1 \times 1$ convolutions with normalization layers to integrate information while maintaining spatial coherence. This fusion approach enables the model to capture complex spatiotemporal patterns essential for accurate forecasting, without compromising computational efficiency.

### 3.2.2 Prediction Module

Building upon the extracted features, our prediction process involves a sequence of specialized components designed to model oceanic dynamics across multiple temporal scales, as illustrated in Figure 1(b).

The prediction module processes the fused representations to model the nonlinear evolution of oceanic states, which consists of stacked attention blocks[28] paired with feed-forward networks. Each attention block captures dependencies across spatiotemporal features, while adaptive layer normalization (AdaLN)[18] incorporates contextual information $\mathbf{F}_S$ to modulate normalization parameters. This design allows the model to focus selectively on relevant oceanic patterns while maintaining computational efficiency.

For feature reconstruction, we employ a shared decoder $\mathbf{D}$ that transforms latent representations back into the physical variable space. The decoder consists of transposed convolutional layers with normalization operations. We establish skip connections between encoder features and corresponding decoder layers, preserving fine-grained spatial details often lost during predictiona critical factor when modeling mesoscale ocean dynamics.

### 3.2.3 Mixture-of-Time Module

A key innovation in our framework is the Mixture-of-Time (MoT) module, illustrated in Figure 1(c), that adaptively integrates predictions from multiple temporal windows. This module addresses a fundamental challenge in ocean forecastingdifferent oceanic variables exhibit distinct temporal evolution characteristics, from fast-changing surface processes to the relatively slow and stable subsurface variability extending down to the twilight zone ( 1500 m). The MoT module performs channel-wise adaptive selection across predictions derived from different temporal windows. Specifically, we maintain a selection matrix $V \in \mathbb{R}^{C \times 4}$ that quantifies prediction reliability across temporal scales for each channel (variable and depth combination). For a given channel $c$ and spatial location $(h, w)$, the final prediction is computed as:

$$\widehat{\mathbf{X}}^{t+1}(c, h, w) = \frac{1}{K} \sum_{i=0}^{3} \text{TopK}_{\mathbf{V}(c,i)} \cdot \overline{\mathbf{X}}_{t-i}^{t+1}(c, h, w) \tag{3}$$

Here, the TopK operator selects the $K$ most reliable temporal windows (corresponding to smallest values in V) for each channel. Specifically, for each channel $c$, it assigns a value of 1 to the $K$ smallest elements in the vector $\mathbf{V}(c, \cdot)$ and 0 to all others. $\overline{\mathbf{X}}_{t-i}^{t+1}$ represents the forecast for time $t + 1$ generated using historical context beginning at time $t - i$. This approach enables variable-specific temporal context selection without increasing model complexity. During training, we update the selection matrix based on prediction performance:

$$\mathbf{V}(c, i) = \alpha \cdot \mathbf{V}(c, i) + (1 - \alpha) \cdot \text{softmax}(\text{AvgPool}_{\text{H,W}}(\text{MAE}(\overline{\mathbf{X}}_{t-i}^{t+1}, \mathbf{X}^{t+1})))_c \tag{4}$$

where $\alpha$ controls update momentum, and MAE measures prediction error. The spatially averaged MAE captures the reliability of each temporal window for each variable, which is then normalized through softmax to ensure the selection weights sum to one. This adaptive approach enables FuXi-Ocean to dynamically adjust its reliance on different temporal windows for each variable and region. For example, deeper ocean current predictions might benefit from recent observations that capture immediate flow evolution, while surface temperature predictions might rely more on longer temporal contexts that effectively represent diurnal cycles and day-night transitions.

### 3.3 Training Strategy

Training deep learning models for ocean forecasting presents unique challenges due to Earth's spherical geometry. To address the varying grid cell areas across latitudes, we implement a latitude-weighted Charbonnier loss[7]:

$$\mathcal{L}_{\text{pred}} = \frac{1}{C \times H \times W} \sum_{c=1}^{C} \sum_{i=1}^{H} \sum_{j=1}^{W} \alpha_i \sqrt{(\widehat{X}_{c,i,j}^{t+1} - X_{c,i,j}^{t+1})^2 + \epsilon^2} \tag{5}$$

where $\alpha_i = H \times \frac{\cos \Phi_i}{\sum_{i=1}^{H} \cos \Phi_i}$ is the latitude-dependent weighting factor at latitude $\Phi_i$, and $\epsilon$ is a small constant ensuring differentiability. This approach prevents bias toward high-latitude regions in error calculations, which would otherwise be overrepresented in the rectangular grid projection.

Besides, we employ a two-stage training approach to enhance model stability and performance. In the first stage, we pre-train the model using a single-step prediction objective. In the second stage, we fine-tune the model with a multi-step loss that penalizes predictions across multiple consecutive time steps, mitigating error accumulation in autoregressive forecasting. This approach follows recent advances in autoregressive modeling [10, 11], which demonstrate the effectiveness of multi-step training for improving long-term forecast stability.

## 4 Experiment

### 4.1 Data

**HYCOM-RD.** We train and evaluate FuXi-Ocean using the HYCOM Reanalysis Data [8], the only publicly available ocean dataset with 6-hour temporal resolution. This dataset features $1/12°$ horizontal resolution with up to 40 vertical layers from sea surface to 5000m depth. We selecte approximately 8.5 years (January 2006 to June 2014) of data for training, a six-month period (July to December 2014) for validation, and one year (January 2015 to December 2015) for testing, focusing on 20 strategically chosen vertical layers down to 1500 m that capture essential ocean dynamics.

**IV-TT Framework.** For evaluation against real-world observations, we employ the GODAE Ocean View Intercomparison and Validation Task Team (IV-TT) Class 4 framework [39], which is obtained from publicly available sources. Details are in the appendix. This framework provides observational datasets from drifting buoys for 2022, alongside interpolated outputs from operational numerical forecasting systems. In addition, we apply a filter to remove anomalous points. Specifically, we compare HYCOM reanalysis data with observational data and remove outliers with high MAE from the observations to ensure fairness in the comparison process. However, after evaluation, the raw data errors for salinity and temperature were too large to be used directly. We primarily evaluate sea surface temperature (SST) forecasts, as this variable exhibits high variability and effectively demonstrates forecasting skill. To ensure consistent comparison, we transform the irregularly distributed observational data to a regular grid matching FuXi-Ocean's output format using nearest-neighbor interpolation. For operational evaluation, FuXi-Ocean is initialized with six-hourly analysis fields from HYCOM.

### 4.2 Implementation Details

The FuXi-Ocean employs the PyTorch framework [35] with the AdamW [23, 29] optimizer, configured with $\beta_1 = 0.9$, $\beta_2 = 0.95$, and a cosine annealing learning rate schedule [30] that decays from $2.5 \times 10^{-4}$ to $10^{-8}$. We train on a cluster of 4 NVIDIA H100 GPUs for 60,000 iterations with a batch size of 1 per GPU, requiring approximately 81 hours to complete. Similar to other autoregressive models [10, 11], FuXi adopts an autoregressive approach during inference, feeding the output back into the input to iteratively generate results for the next time step. For further details, please refer to the supplementary materials.

### 4.3 Metrics

Following standard practices in operational ocean forecast evaluation[11], we use latitude-weighted root mean square error (RMSE) as our primary evaluation metrics to account for the varying grid cell

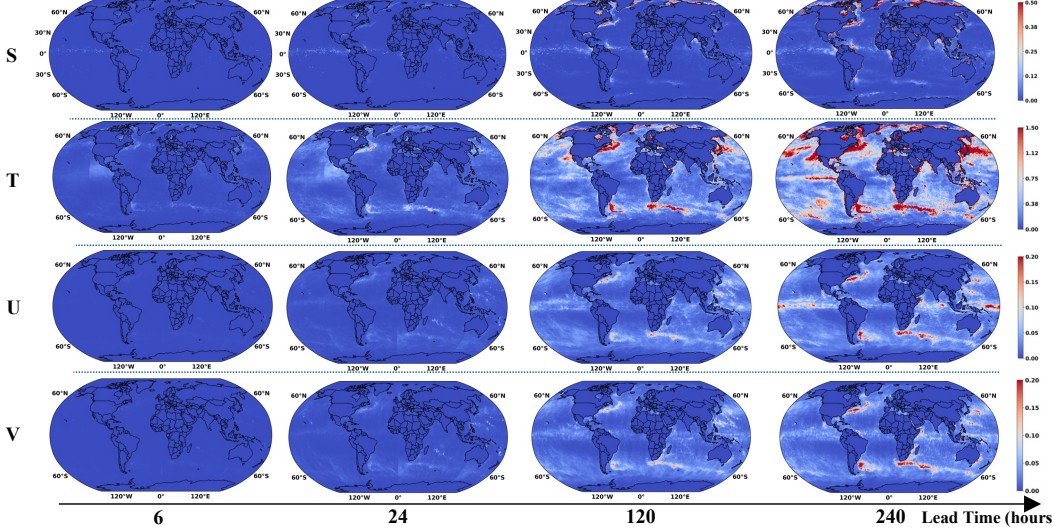

Figure 2: **Global RMSE distribution of sea surface.** From top to bottom, the results represent salinity (psu), temperature (řC), and ocean current U/V components (m/s), and from left to right correspond to different forecast lead times. Each subplot represents the average RMSE (lower is better) for the test set.

sizes across different latitudes, formulated as follows:

$$\text{RMSE}(c, \tau) = \frac{1}{|\text{D}|} \sum_{t_0 \in \text{D}} \sqrt{\frac{1}{\text{H} \times \text{W}} \sum_{i=1}^{\text{H}} \sum_{j=1}^{\text{W}} a_i (\widehat{\mathbf{X}}_{c,i,j}^{t_0+\tau} - \mathbf{X}_{c,i,j}^{t_0+\tau})^2} \tag{6}$$

where $t_0$ denotes the forecast initialization time in the testing dataset D, and $\tau$ represents the lead time steps added to $t_0$.

## 5 Results

We first evaluate FuXi-Ocean on the 2015 HYCOM set derived from HYCOM reanalysis data to assess its ability to generate high-frequency (6-hourly) predictions. We then compare our model with operational forecasting systems using 2022 observational data through the GODAE Ocean View IV-TT framework. This dual evaluation approach allows us to comprehensively assess both the model's intrinsic predictive capabilities and its real-world operational performance relative to established systems.

### 5.1 Performance testing

**Horizontal Spatial Error Analysis.** Fig. 2 shows the global spatial distribution of RMSE for sea surface variables (temperature, salinity, and currents) at different lead times (6, 24, 120, and 240 hours). For temperature (T), the model maintains particularly low errors in the tropical and subtropical regions across all forecast horizons, with RMSE values typically below 0.3řC for 6-hour predictions. Error patterns intensify primarily in western boundary current regions (Gulf Stream, Kuroshio Current) and the Antarctic Circumpolar Current, where intense mesoscale eddy activity creates inherent predictability challenges [31, 12]. Notably, due to the high uncertainty of changes in these regions, even observational data from satellites and other sources can be significantly affected, leading to instability in model results [32, 12]. Salinity (S) forecasts display a similar spatial pattern but with larger relative errors in regions of significant freshwater influence, such as major river outflows and high-precipitation zones [25]. The equatorial Pacific shows particularly strong performance across all lead times, maintaining low error values even at 10-day forecasts. At shorter forecast times, the error distribution locations of the current velocity components U and V are highly similar. As the forecast lead time increases, significant differences in U and V near the equator

emerge, which aligns with the physical phenomenon of the inconsistent rotation directions of warm currents in the Northern and Southern Hemispheres. Western boundary currents show the highest error magnitudes, consistent with their intrinsic variability. Nevertheless, FuXi-Ocean maintains reasonable accuracy in these challenging regions, with errors remaining within operational tolerance thresholds even at longer lead times. Crucially, our 6-hour predictions show substantially lower errors than daily forecasts across all variables, validating the model's ability to capture sub-daily ocean dynamics.

**Vertical Performance Analysis.** The vertical profile of the ocean environment is a crucial aspect for

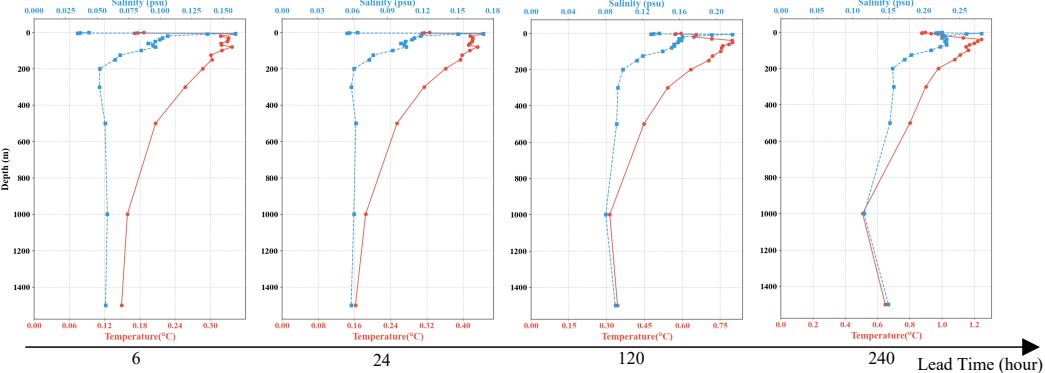

Figure 3: **Depth-dependent RMSE Distributions of salinity and temperature varying with lead time.** Each subplot represents the RMSE (lower is better) varying with depth at the current lead time. Blue represents salinity results, while red represents temperature results.

evaluating the effectiveness of ocean prediction systems. In Fig. 3, we present the variation of RMSE for salinity and temperature with depth. First, FuXi-Ocean maintains consistent prediction skill throughout the water column, with RMSE degrading gradually with depth. Second, the thermocline region (approximately 100-300m) shows relatively higher errors compared to both surface and deep layers, reflecting the inherent difficulty in predicting this dynamically complex interface. Despite this challenge, FuXi-Ocean's RMSE still maintains effective values. Additionally, the error growth with increasing forecast lead time shows an interesting depth-dependent pattern. Thanks to our Mixture-of-Time (MoT) approach, which adaptively weights temporal dependencies for each variable, even though surface layers (0-300m) are sensitive to rapidly changing conditions while deeper layers (beyond 500m) are not, the error growth across layers remains relatively stable without any layer performing significantly worse.

## 5.2 Comparison

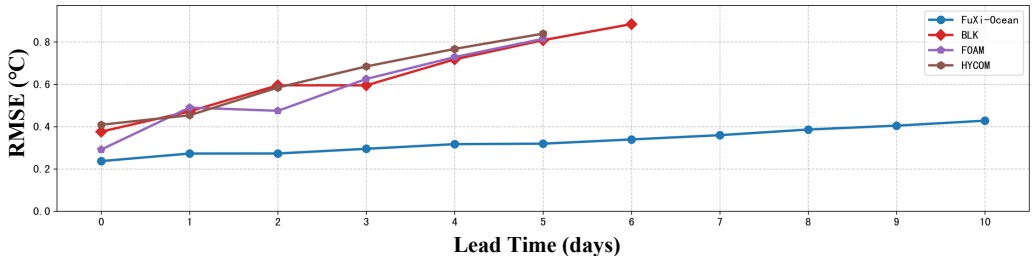

Figure 4: **The SST comparison of different methods based on the IV-TT evaluation framework.** The x-axis represents the forecast time, and the y-axis represents the RMSE (lower is better).

To evaluate the actual operational performance, we used sea surface temperature observation data from 2022 within the IV-TT framework to compare FuXi-Ocean with the HYCOM, BLK, and FOAM methods. For this comparison, we averaged our 6-hourly outputs to produce daily means that can be directly compared with the daily forecasts of other methods. To ensure a fair comparison, we also evaluated the RMSE at the initial time (0 day). As shown in Fig. 4, the RMSE of FuXi-Ocean is

lower than that of other methods, and the cumulative error growth is significantly smaller. Note that the most recent training data for FuXi-Ocean does not exceed 2014, and it uses only oceanic variables as input information, highlighting that FuXi-Ocean effectively captures the intrinsic patterns of ocean system changes.

## 5.3 Ablation

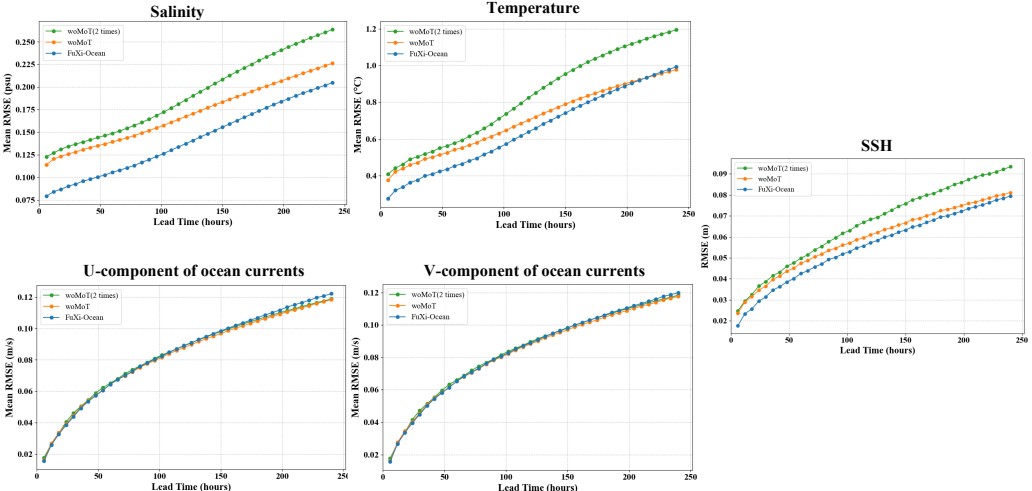

Figure 5: **The ablation study of different methods on the HYCOM-RD.** The five subplots respectively show salinity, temperature, ocean current UV components, and sea surface height. The x-axis represents the forecast lead time, and the y-axis represents the RMSE (lower is better). Note that for each variable, we average the RMSE over depth.

To evaluate the contribution of our key innovations, we compare FuXi-Ocean with two ablated variants: (1) "woMoT," which removes the MoT module, and (2) "woMoT(2times)," which removes MoT and reduces input time steps from four to two.

Figure 5 presents the results across all five forecasted variables. For salinity, temperature, and SSH, removing the MoT module leads to substantial performance degradation, with the increasing of RMSE peaked at nearly 40% across forecast lead times. This degradation is particularly pronounced for short-term forecasts, where adaptive temporal context selection proves critical for capturing rapidly evolving processes. These findings confirm that variable-specific temporal modeling significantly improves prediction accuracy for thermohaline variables that exhibit complex spatiotemporal dynamics. Furthermore, reducing the historical context from four to two time steps (woMoT(2times)) causes additional performance deterioration, especially for temperature and salinity predictions beyond 48 hours. This contrasts with experiences in atmospheric forecasting models, where shorter historical windows often suffice. Interestingly, ocean current components (U and V) show less sensitivity to both the removal of MoT and the reduction of temporal context. This observation aligns with the physical reality that surface currents are largely driven by recent wind forcing and geostrophic balance rather than their own history. The minimal performance difference between ablated models for current forecasting suggests that specialized architectures may be warranted for different ocean variablesa direction we leave for future work.

## 6  Conclusion

We present FuXi-Ocean, the first data-driven global ocean forecasting system achieving 6-hour temporal resolution at a $1/12°$ spatial scale with coverage from surface to 1500 m depth. Our Mixture-of-Time module adaptively captures different temporal dependencies across ocean variables, addressing a key limitation of previous models. FuXi-Ocean outperforms operational numerical systems in predicting sea surface temperature while requiring significantly fewer computational resources and relying only on ocean variables as input. Its 6-hour forecasts demonstrate superior

accuracy than daily-averaged predictions from state-of-the-art numerical models, confirming the effectiveness of our approach for high-frequency ocean prediction.

The successful application of deep learning to sub-daily ocean forecasting opens several promising avenues for future research directions. First, while our current model covers depths up to 1500 m, extending its range to abyssal depths would provide a more complete representation of the ocean system. Second, incorporating physical conservation laws as additional constraints could further improve prediction stability and ensure physical consistency. Third, increasing the forecast frequency beyond 6 hours toward hourly predictions could enable applications requiring even finer temporal resolution, such as tidal forecasting and coastal hazard warning systems.

## Acknowledgments and Disclosure of Funding

This work was supported by AI for Science Program (ID: 2025-GZL-RGZN-BTBX-02017), Shanghai Municipal Commission of Economy and Information

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

# A    Limitations

Despite these promising results, our approach has several limitations that warrant further investigation. The reliance on reanalysis data for training, while necessary given data availability constraints, introduces potential biases from the underlying numerical models. This challenge is compounded by the scarcity of publicly available high-quality sub-daily ocean reanalysis data. In future work, we plan to address this limitation by integrating HYCOM analysis fields with Argo observational data and other in-situ measurements to create a more robust dataset. For subsurface validation, this will involve a rigorous data matching process, where observational profiles are spatio-temporally collocated with the model grid points, and quality control filters are applied to remove outliers and inconsistent measurements before comparison. The filters use HYCOM analysis field data as an anchor to remove outlier data with high variability. Additionally, our evaluation thus far focus primarily on standard RMSE metrics, which, while useful for comparing forecast systems, may not fully capture the model's ability to represent specific ocean phenomena like mesoscale eddies, western boundary currents, or seasonal thermocline transitions. Future analysis will incorporate phenomenon-specific evaluation metrics and physical consistency measures to provide deeper insights for model improvement. Finally, while our system's forecast accuracy remains robust through 10 days, performance for longer-term predictions (seasonal to interannual) remains unexplored, representing another important direction for future research.

# B    Broader Impacts

FuXi-Ocean represents a significant advancement with far-reaching implications across multiple domains. By enabling high-resolution, sub-daily ocean forecasting at global scales, our work creates opportunities for numerous scientific and societal applications. In the realm of maritime safety and operations, the 6-hour temporal resolution provides critical advantages for shipping navigation, offshore energy production, and search and rescue missions. Particularly for emergency response scenarios such as oil spill tracking and marine accident response, the ability to forecast ocean currents at 6-hour intervals significantly improves source identification and trajectory prediction, potentially saving lives and reducing environmental damage. For marine resource management, FuXi-Ocean can enhance fisheries operations through improved forecasting of ocean conditions that influence fish migration and aggregation patterns. The comprehensive vertical coverage (0-1500 m) is especially valuable for this application, as it captures the habitats of commercially important species that undergo diel vertical migration. For coastal communities and small island nations, FuXi-Ocean provides enhanced capability for predicting coastal hazards like storm surge, coastal flooding, and harmful algal blooms, all of which can develop rapidly and require high-frequency forecasting systems for adequate warning. The eddy-resolving capabilities of our model are particularly important for these applications, as smaller-scale coastal processes often drive the most damaging impacts. From a computational efficiency perspective, FuXi-Ocean demonstrates that high-quality forecasts can be generated with significantly reduced computational resources compared to traditional numerical models. This efficiency makes sophisticated ocean forecasting more accessible to researchers and institutions with limited computational infrastructure, potentially democratizing access to advanced oceanographic tools.

However, our work also presents potential risks that warrant careful consideration. Relying on deep learning models for critical ocean forecasting applications requires robust verification systems, particularly when extending predictions to new or unprecedented scenarios not represented in the training data. The reliance on reanalysis data means that biases in the underlying numerical models could be perpetuated or even amplified by our approach. Additionally, as with any forecasting system, there is risk in over-reliance on model outputs for critical decision-making without appropriate uncertainty quantification.

# C    Additional results

## C.1    Performance Test Supplement

We present our test results for ocean current UV in Fig.6 and 7. Similar to Fig.3 in the main text, the RMSE here is tested on the HYCOM reanalysis data for 2015. From the results, it can be seen that

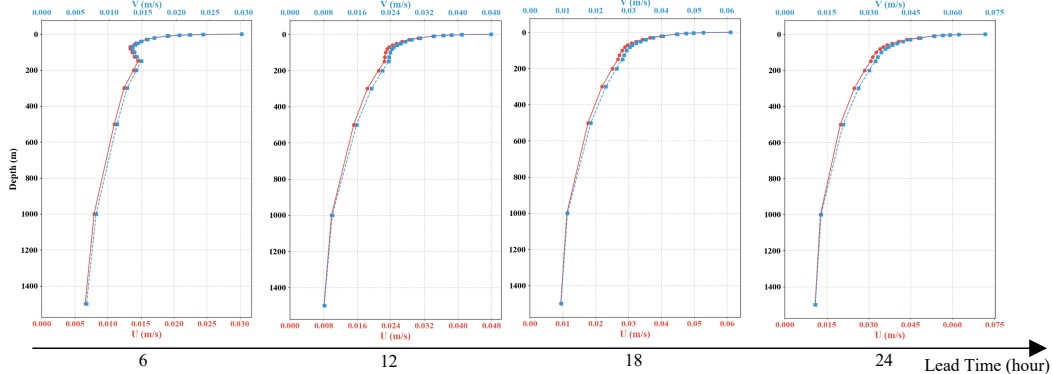

Figure 6: **Depth-dependent RMSE Distributions of ocean current UV components varying with lead time (6 - 24 hours).** Each subplot represents the RMSE (lower is better) varying with depth at the current lead time.

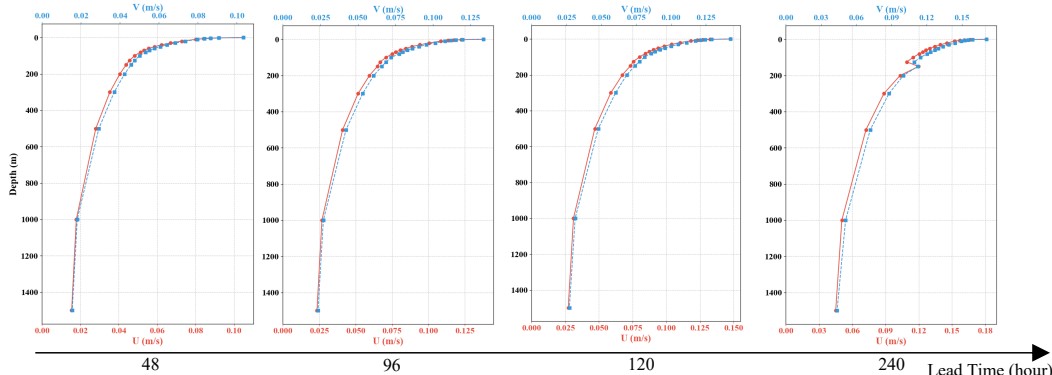

Figure 7: **Depth-dependent RMSE Distributions of ocean current UV components varying with lead time (48 - 240 hours).** Each subplot represents the RMSE (lower is better) varying with depth at the current lead time.

the RMSE values for ocean current UV maintain high accuracy and exhibit high stability in error accumulation. As lead time increases, the error growth is gradual and uniform, without significant differences between intra-day and daily intervals. Notably, in the 6-hour forecast, the stratification of change characteristics near the ocean surface is evident, indicating that the changes in ocean currents are more easily influenced by recent changes, such as sea surface winds.

### C.2    Performance Test Supplement

To comprehensively evaluate the model's performance, we conducted tests on ocean current UV components under the observation data (Buoy data on CMEMS). The WenHai* and Glory* shown in the fig.8 and 9 are direct comparisons based on descriptions from paper[11]. Although this comparison may not be entirely fair (mainly due to inconsistent testing years), FuXi-Ocean's performance can still serve as a simple reference.

## D    Data

As shown in the Tab.1, compared with the long-term training data used by mainstream ocean models such as Xihe, Wenhai, Langya, etc., we used shorter time range data for training and achieved better prediction results. For data-driven deep learning methods, we can accurately capture ocean phenomena and features by relying solely on pure ocean variables such as T, S, U, V and SSH. **Observation Data.** For the IV-TT data, we list the available public links in Fig.2, which include salinity, temperature, and SST variables. Through experimentation, it was found that due to practical

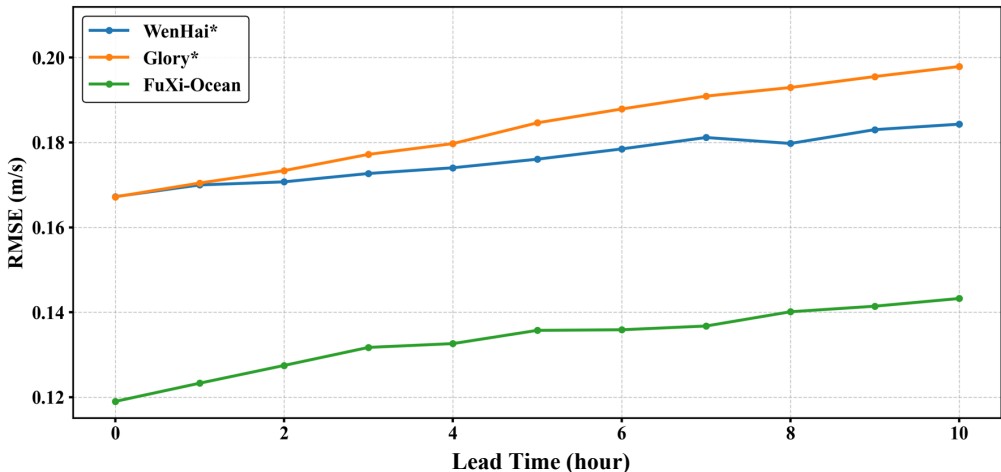

Figure 8: **The ocean current U comparison of different methods based on the observation data.**
The x-axis represents the forecast time, and the y-axis represents the RMSE (lower is better). Note
that the results of WenHai and Glory are derived from their papers, hence marked with an asterisk (*).

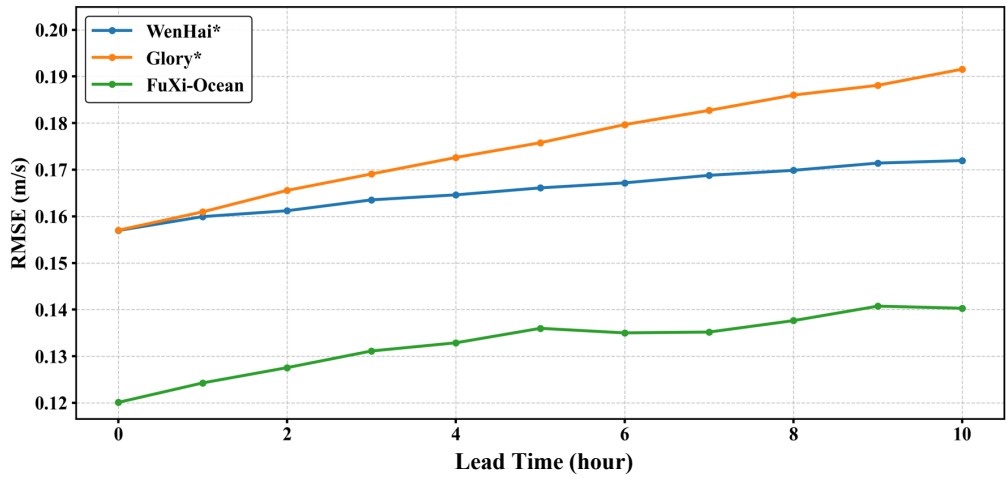

Figure 9: **The ocean current V comparison of different methods based on the observation data.**
The x-axis represents the forecast time, and the y-axis represents the RMSE (lower is better). Note
that the results of WenHai and Glory are derived from their papers, hence marked with an asterisk (*).

issues, only partial data is available. Firstly, due to the presence of outliers in the temperature and
salinity vertical profile data from IV-TT, these anomalies significantly impact the evaluation results,
causing the errors to be abnormally large. Secondly, because the ocean contains thermoclines and
haloclines, where temperature and salinity change rapidly and non-linearly, using IV-TT to perform
linear interpolation in the vertical direction to align with the model grid introduces excessive errors,
thereby reducing the reliability of the evaluation results. Therefore, we ultimately use SST for

Table 1: Comparison of Training Data for Existing Ocean Models

| Name of the Model | Training Data Time Range | Input Variables |
|---|---|---|
| Xihe | 1993-2020 | SST, U10, V10, T, S, U, V, SSH |
| Langya | 1993-2021 | SST, U10, V10, T, S, U, V, SSH |
| WenHai | 1993-2020 | SST, U10, V10, T, S, U, V, SSH |
| FuXi-Ocean | 2006-2015 | T, S, U, V, SSH |

Table 2: The source of observation data

| Variable | Data type | Source of data |
|---|---|---|
| T | In situ Argo profiles | Argo GDAC (from `http://www.usgodae.org/argo/argo.html` or `http://www.coriolis.eu.org/`) |
| S | In situ Argo profiles | Argo GDAC (as above) |
| U/V | In situ Argo profiles | Argo, Ocean Sites, GOSUD, EGO `https://data.marine.copernicus.eu/product/INSITU_GLO_PHYBGCWAV_DISCRETE_MYNRT_013_030` |
| SLA | Satellite altimeter from Jason-1, Jason-2 and Envisat | CLS Aviso Level 3 data |
| SST | In situ surface drifter data | From USGODAE server: `http://www.usgodae.org/cgi-bin/datalist.pl?dset=fnmoc_obs_sfcobs&summary=Go` |

evaluation and comparison.

The observational data from ARGO is not gridded, and its degree of discreteness is shown in Fig.10. Globally, the distribution of stations at key locations is relatively uniform and comprehensive.

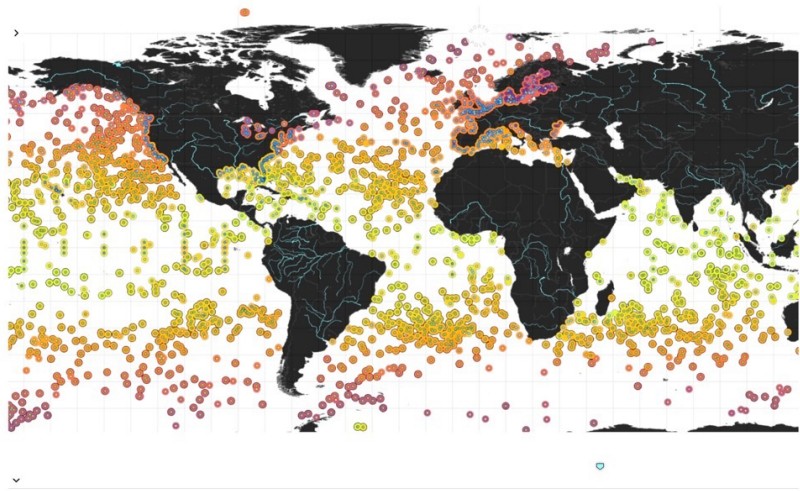

Figure 10: **Global distribution map of ARGO and other buoys.**

# E  Supplementary details

In practice, we set $K = 1$ to achieve the best performance of the MoT module. In the module design, we assume that there will always be an optimal moment of information that can optimize the results. Of course, if the number of historical moments involved, $N$, increases significantly, $K = 1$ is unlikely to be a good choice.

