# OpenReview forum: "FuXi-Ocean: A Global Ocean Forecasting System with Sub-Daily Resolution"
_NeurIPS.cc/2025/Conference — NeurIPS 2025 oral_

### Official Review · Reviewer_sFCf · 2025-06-30

**Clarity:** 3
**Significance:** 2
**Originality:** 3
**Rating:** 4
**Confidence:** 4

**Summary:**

In this paper, the authors introduce FuXi-Ocean, a new data-driven global ocean forecasting model that delivers six-hourly predictions at an eddy-resolving 1/12° spatial resolution—offering a big step up in temporal detail compared to earlier approaches. What’s more, while many previous models only go down to about 700 meters, FuXi-Ocean dives much deeper, reaching 1500 meters to capture complex ocean dynamics like thermocline shifts and deep-water mass behavior.
One of the standout features is its Mixture-of-Time (MoT) module, which smartly learns how reliable different variables are across various time scales. Experimental results demonstrate that FuXi-Ocean can achieve state-of-the-art accuracy across key variables like temperature, salinity, ocean currents, and sea surface height.

**Questions:**

To further solidify the novelty and impact of the Mixture-of-Time (MoT) module , I recommend adding a dedicated discussion comparing MoT with temporal attention mechanisms , which are commonly used in sequence modeling (e.g., Transformers in weather prediction). This comparison would help readers better understand the unique advantages of MoT in the context of ocean forecasting.

**Ethical Concerns:**

["NO or VERY MINOR ethics concerns only"]

**Final Justification:**

Thank you to the authors for their time and detailed responses during the rebuttal. After reviewing all the comments, I would like to raise my score.

**Limitations:**

Yes

**Quality:**

3

**Strengths And Weaknesses:**

This paper presents a well-structured deep learning framework, FuXi-Ocean, featuring novel architectural components such as the Mixture-of-Time (MoT) module and context-aware feature extraction. The authors conduct a fairly comprehensive evaluation using both HYCOM reanalysis data and real-world observational datasets (via the IV-TT framework). Comparisons with operational numerical models (HYCOM, BLK, FOAM) and other data-driven approaches (XiHe, Langya, WenHai) highlight FuXi-Ocean’s strong performance.

However, I remain somewhat unconvinced about the effectiveness of the MoT module. As shown in Figure 5, for the U and V components of ocean currents, there appears to be little to no performance gain when MoT is applied. This suggests that MoT may not be the right solution for these variables—at least in its current form. As the authors themselves note, future work should consider designing specialized modules that can better capture inter-variable interactions, rather than relying solely on historical temporal patterns.

---

> ### Author Rebuttal · Authors · 2025-07-31
>
> Dear Reviewer sFCf,
>
> Thank you for your comments and suggestions on our work. We appreciate your recognition of the novel contributions of our work and are grateful for the opportunity to address your concerns.
>
> ## The effectiveness of the MoT module.
>
> A: We appreciate your interest in understanding the effectiveness of the **Mixture-of-Time (MoT)** module in ocean sub-daily forecasting models. To evaluate its performance, we replace the MoT module with temporal attention mechanisms, which force the model to consider the influence of all historical temporal features for each variable. We test on HYCOM 2015 reanalysis data, and average the results across the depth dimension for each variable. While we cannot provide intuitive line graphs to illustrate the comparison, we present the **normalized RMSE** to highlight the impact of the MoT module on forecasting accuracy.
>
> The normalized RMSE is calculated as:
>
> $\frac{RMSE - RMSE_{FuXi-Ocean}}{RMSE_{FuXi-Ocean}} *100\%$
>
> The results are shown in the table below:
> | Time | S     | T    | U    | V    | SEL   |
> |------|-------|-------|-------|-------|-------|
> | 6    | 0.897 | 1.103 | 0.512 | 0.394 | 1.108 |
> | 24   | 2.486 | 1.821 | 0.795 | 0.713 | 1.194 |
> | 54   | 1.089 | 1.217 | 0.506 | 0.491 | 1.115 |
> | 72   | 1.412 | 1.185 | 0.708 | 0.597 | 1.287 |
> | 120  | 1.193 | 1.408 | 0.403 | 0.296 | 1.206 |
> | 240  | 1.507 | 1.592 | 0.694 | 0.815 | 1.389 |
>
> It is evident from the table that while the model was provided with all temporal features, **more feature information does not always lead to better performance**, particularly in the case of **ocean sub-daily forecasting tasks**. The performance of the model significantly deteriorates at **24-hour forecasts**, suggesting that the inclusion of more temporal features **interferes with the learning of diurnal cycle characteristics**. This is an important observation that highlights the challenge of balancing the complexity of temporal information in sub-daily ocean forecasting models.
>
> ## The U and V components of ocean currents.
>
> A: As we mentioned in our paper, while the MoT module has shown **significant improvements** for variables with **diurnal cycles** (e.g., temperature and salinity), its application to **ocean currents** presents a different set of challenges due to the **slower timescales** of these variables. We acknowledge that **U and V currents** are more **complex** and may require additional design considerations. However, the model’s limitations in this regard do not detract from the broader impact of our work.
>
> Our objective was not to fully solve the challenge of predicting ocean currents but to **introduce the first global ocean forecasting model with sub-daily resolution**. This was the primary aim of our paper. The improvements we’ve achieved for surface variables with diurnal cycles demonstrate that FuXi-Ocean **paves the way** for future advancements in this field. We believe that **specialized techniques for ocean currents** can be developed as a natural extension of our work, and we plan to explore this in future research.

---

### Official Review · Reviewer_7qdU · 2025-07-02

**Clarity:** 4
**Significance:** 4
**Originality:** 2
**Rating:** 5
**Confidence:** 4

**Summary:**

The paper proposes an autoregressive machine learning (ML) emulator of the surface ocean, including state variables up to 1500m depth, at 1/12° resolution, and six-hourly time steps. The proposed method is based on an encoder-processor-decoder framework with a newly proposed "mixture-of-time" component that is intended to modulate the temporal context for each prognostic variable. The method is trained on reanalysis and evaluated against the reanalysis and SSTs from an observational dataset using RMSE.

**Questions:**

- How does the method improve on existing methods in the comparison on reanalysis?
- Why does the model not use atmospheric forcings as input?
- What is the maximum temporal history of the model?
- Is Figure 4 a fair comparison?

**Ethical Concerns:**

["NO or VERY MINOR ethics concerns only"]

**Final Justification:**

The authors have addressed all my questions and exceeded expectations in doing so. The paper proposes a global ocean emulator that outperforms a persistence baseline, which is much less trivial than it may sound. This ocean emulator will be useful for next steps towards seasonal and climate emulators.

**Limitations:**

- Several limitations were left unclear. It help to discuss limitations regarding blurring due to RMSE, lack of surface forcings, maximum timescales, and observational errors in temperature and salinity.
- The ethical concerns are well written. It would have been helpful to mention the risks of overfishing in the sentence on using this model for guiding large-scale fishing vessels.

**Paper Formatting Concerns:**

-

**Quality:**

3

**Strengths And Weaknesses:**

Strengths:
- The work is very well motivated and ML ocean emulators are certainly a key topic of interest in the research community. Ocean emulators are necessary for subseasonal to seasonal forecasting and coupled ocean-atmosphere clime model emulators.
- The choice to use more than two previous steps as inputs is well motivated for the ocean. Other work has shown that the time-step size and history are important hyperparameters (e.g., https://doi.org/10.22541/essoar.170110658.85641696/v1) and it is helpful to see the authors spend a significant amount of effort thinking about this.
- The dataset, experiment setup, and evaluation metrics are very clearly described making is easy to understand the results section.

Weaknesses
- Most importantly the paper needs a baseline for Figure 2 and 3. It is entirely unclear to me if these results are improving on existing methods, and it is hard to tell if e.g., 0.5 salinity is a high or low error. I would ask the authors to add a 'persistence' baseline, i.e., a forecast that assumes the state at t=0 to be constant throughout the prediction window,  into the paper. I would also recommend to add a climatology baseline, and - I don't think it's required - but the authors would greatly improve the impact of the paper if they were to add HYCOM, BLK, or FOAM hindcasts into the comparison vs reanalysis.
- Why does this forecasting problem work if there are no forcings provided from the atmosphere? As far as I understand, SST, salinity, U and V at the ocean surface layer are largely influenced by atmospheric winds, temperatures, and rainfall. I would have assumed that the model should take these forcings as inputs, but they don't seem to be used. Could the authors please detail why these forcings are not necessary?
- The arguments on "long-term deep ocean memory" and "slow-evolving deep-water phenom­ena" seemed a bit of a stretch to me. Does the model have the capability to leverage information further than 24hrs, i.e., four time steps ago? There are deep ocean processes that require history windows of over 50 years and the current model architecture doesn't seem like it would capture those. Thus, I recommend to rephrase those sections in the paper and clarify the timescales that the paper can pick up on.

Minor comments:
- Can the authors change the wording from "deep ocean" into "twilight zone" or "deeper ocean". I find the denomination deep ocean quite misleading as it typically includes all depth beyond 200m, incorrectly implying that FuXi-Ocean would contain state variables from the abyssal and benthic zone.
- 3.2.3 was a bit hard to understand for me. For example how is a "temporal window" defined here? Given that the MoT layer is such an important piece of this paper it would have helped to expand a bit on this layer in the appendix. For example, are there similar approaches in other papers? What are the limitations of this layer? How is K determined and why did the authors choose K=1?
- There a sentence in the introduction: "designing [...] models requires [...] while avoiding overfitting to spurious correlations". This is very interesting, as I believe it is a common issue of ML autoregressive emulators that they do not learn correct sensitivities to each input variables. It is not necessary, but would make this paper excellent if the authors included a study to test if the model has learned correct sensititivites to each input variable.
- The paper includes a sentence "after evaluation, the raw data errors for salinity and temperature were too large to be used directly". There is detail provided on this in the appendix, but it would be helpful to have one additional sentence on this in the main text.
- Figure 2 plot is missing units
- Un Fig 4, what are the references for HYCOM, BLK, and FOAM methods. Is HYCOM in this case a reanalysis or a forecast? If it's a reanalysis than there is likely an issue in the put because a forecast almost never outperforms a reanalysis.
- The results in Figure 8 and 9 seem suspiciously good. Could the authors please detail in which ways this is NOT a fair comparison? In particular I would have expected Glory and WenHai to have better RMSE scores due to their access to U10 and V10 wind forcing fields. Do I understand correctly that they're using UV10 as forcing and not as prognostic variables? I would probably recommend
- Evaluation on RMSE only seems quite limiting. It would significantly increase the impact of this paper if the evaluation of spectra were included. It's unclear if the model predicts blurred fields.
- Typos:
resolutions deep learning unique challenges.
preivous models
and relying only ocean variables as input
Implement Details
B BoraderImpacts

I am willing to raise my score if the proposed changes are adequately addressed.

---

> ### Author Rebuttal · Authors · 2025-07-31
>
> Dear Reviewer 7qdU,
>
> Thank you for your insightful review and constructive feedback. We appreciate your recognition of our work's strong motivation and clear experimental setup. Your expertise in ocean dynamics is evident from your thoughtful questions about atmospheric forcing and temporal memory. We have conducted additional experiments and analyses to address all your concerns comprehensively.
>
> ## Q1. Compare with baseline.
>
> A: We have added persistence baseline comparisons as requested. Following WenHai's approach, we present results on observational data for sea surface variables to avoid introducing HYCOM data biases. The baseline uses HYCOM initial fields to calculate persistence RMSE.
>
> |Method|Variable|Day 0|Day 1|Day 2|Day 3|Day 4|Day 5|Day 6|Day 7|Day 8|Day 9|Day 10|
> |-------------|----------|--------|--------|--------|--------|--------|--------|--------|--------|--------|--------|--------|
> |Persistence|S0|0.110|0.115|0.116|0.121|0.124|0.123|0.127|0.130|0.135|0.136|0.139|
> ||T0|0.236|0.274|0.310|0.356|0.396|0.438|0.487|0.522|0.573|0.615|0.664|
> ||U0|0.119|0.145|0.154|0.156|0.155|0.157|0.159|0.161|0.162|0.161|0.163|
> ||V0|0.120|0.146|0.154|0.156|0.157|0.159|0.159|0.161|0.163|0.161|0.162|
> |FuXi-Ocean|S0|0.110|0.113|0.115|0.118|0.119|0.121|0.118|0.123|0.122|0.125|0.126|
> ||T0|0.236|0.272|0.277|0.295|0.317|0.319|0.339|0.359|0.386|0.404|0.428|
> ||U0|0.119|0.123|0.127|0.132|0.133|0.136|0.136|0.137|0.140|0.141|0.143|
> ||V0|0.120|0.124|0.128|0.131|0.133|0.136|0.135|0.135|0.138|0.141|0.140|
>
> *Units: S0 (PSU), T0 (°C), U0/V0 (m/s)*
> The results clearly demonstrate that FuXi-Ocean consistently outperforms persistence for all variables, with the advantage increasing over time.
>
> ## Q2. Atmospheric forcings.
>
> A: We strongly agree that atmospheric forcing is crucial for ocean forecasting. To quantify this, we conducted experiments incorporating five atmospheric variables from ERA5 (10-m zonal/meridional winds, mean sea level pressure, 2-m temperature, and 2-m dewpoint temperature) using an additional encoder network for atmosphere-ocean fusion.
>
> The table shows relative RMSE improvement when adding atmospheric forcing: $\frac{\text{RMSE}_{with\_atm} - \text{RMSE}_{FuXi-Ocean}}{\text{RMSE}_{FuXi-Ocean}} \times 100\%$ (negative values indicate improvement):
>
> |Lead Time (h)|S|T|U|V|SEL|
> |---------------|---------|---------|---------|---------|---------|
> |6|-2.52%|-4.30%|-1.22%|-1.20%|-1.71%|
> |24|-0.65%|-6.64%|-2.61%|-2.50%|-3.06%|
> |54|-1.19%|-3.88%|-3.56%|-3.14%|-2.22%|
> |72|-3.41%|-5.66%|-3.06%|-2.46%|-2.15%|
> |120|-6.36%|-5.85%|-5.21%|-4.43%|-4.40%|
> |240|-9.00%|-11.05%|-8.06%|-8.85%|-12.82%|
>
> *Units: S (PSU), T (°C), U/V (m/s), SEL (m)*
> Specifically, we conducted comprehensive testing on the 2015 HYCOM reanalysis data and average the results across the depth dimension. As expected, atmospheric forcing improves all variables, particularly temperature at longer lead times. However, our key contribution is demonstrating that effective sub-daily forecasting is achievable using ocean variables alone, with performance exceeding daily numerical models. This approach:
>
> 1. Reduces data dependencies and computational overhead
> 2. Leverages the ocean's inherent predictability from its large heat capacity and momentum
> 3. Provides a strong baseline that can be enhanced with atmospheric coupling
>
> This represents a different paradigm from traditional ocean forecasting—utilizing the ocean's internal dynamics rather than treating it as purely atmospherically driven. Our upcoming work will present a fully coupled ocean-atmosphere system building on this foundation, but the current results establish the surprising effectiveness of ocean-only deep learning approaches for short-term forecasting.
>
> ## Q3. The maximum temporal history of the model.
>
> A: Thank you for pointing out the ambiguity in our terminology. We will revise "temporal windows" to "historical time points" for clarity.
>
> Regarding the temporal extent accessible to FuXi-Ocean: While the architecture could theoretically accommodate longer histories, our design choices are guided by the specific requirements of sub-daily ocean forecasting:
>
> 1. **Physical considerations**: The 6-hour interval with 4 historical time points (24 hours total) captures the complete diurnal cycle—the dominant mode of variability for sub-daily forecasting.
>
> 2. **Practical constraints**: Extending the temporal history significantly increases:
>    - Memory requirements (linear with time points)
>    - Computational cost during training and inference
>    - Data loading overhead in operational settings
>
> 3. **Task-specific design**: FuXi-Ocean targets medium-range (1-10 day) forecasting where 24-hour history provides optimal information content. For seasonal/subseasonal prediction, different architectural choices would be appropriate, as you correctly note.
>
> We acknowledge that processes like deep ocean memory spanning decades exist, but these operate on timescales beyond our forecast horizon and would require fundamentally different modeling approaches.
>
> ## Q4. Sensitivity of K in the MoT module.
>
> A: We conducted extensive experiments with K = 2, 3, 4, and also tested a soft aggregation approach. The table below shows the relative RMSE change compared to K=1, calculated as $\frac{\text{RMSE} - \text{RMSE}_{K=1}}{\text{RMSE}_{K=1}} \times 100\%$. For each variable, we averaged the results across the depth dimension. The "attn" group performs soft aggregation on the four temporal features using attention mechanisms.
>
> |Lead Time (h)|S (K=2)|S (K=3)|S (K=4)|S (attn)|T (K=2)|T (K=3)|T (K=4)|T (attn)|U (K=2)|U (K=3)|U (K=4)|U (attn)|V (K=2)|V (K=3)|V (K=4)|V (attn)|SEL (K=2)|SEL (K=3)|SEL (K=4)|SEL (attn)|
> |-|--|----|---|---|----|---|--|----------|--------|--------|--------|----------|--------|--------|--------|----------|---------|---------|---------|-----------|
> |6|+1.07%|+1.05%|+1.06%|+0.90%|+1.11%|+1.10%|+1.12%|+1.10%|+1.21%|+1.43%|+1.53%|+0.51%|+1.21%|+1.32%|+1.42%|+0.39%|+1.12%|+1.11%|+1.12%|+1.11%|
> |24|+1.00%|+2.32%|+3.00%|+2.49%|+1.18%|+1.60%|+1.78%|+1.82%|+1.10%|+0.93%|+0.95%|+0.80%|+1.10%|+0.92%|+1.01%|+0.71%|+1.21%|+1.52%|+1.67%|+1.19%|
> |54|+1.01%|+1.05%|+1.01%|+1.09%|+1.01%|+1.10%|+1.12%|+1.22%|+1.07%|+1.88%|+1.52%|+0.51%|+1.06%|+1.81%|+1.53%|+0.49%|+0.97%|+1.32%|+1.11%|+1.12%|
> |72|+1.03%|+1.10%|+1.04%|+1.41%|+1.03%|+1.24%|+1.21%|+1.19%|+1.13%|+2.62%|+1.79%|+0.71%|+1.12%|+2.42%|+1.75%|+0.60%|+1.00%|+1.70%|+1.26%|+1.29%|
> |120|+1.09%|+1.35%|+1.14%|+1.19%|+1.10%|+1.79%|+1.42%|+1.41%|+1.48%|+3.94%|+2.40%|+0.40%|+1.47%|+3.44%|+2.30%|+0.30%|+1.06%|+2.64%|+1.55%|+1.21%|
> |240|+1.03%|+1.62%|+1.03%|+1.51%|+1.35%|+2.80%|+1.56%|+1.59%|+2.23%|+5.09%|+2.87%|+0.69%|+2.32%|+4.61%|+2.78%|+0.82%|+1.77%|+3.94%|+2.19%|+1.39%|
>
>
> *Units: S (PSU), T (°C), U/V (m/s), SEL (m)*
>
> Our experiments reveal that K=1 consistently yields optimal performance. The degradation at 24-hour lead time is particularly revealing—it coincides with the diurnal cycle period, supporting our hypothesis that single-context selection prevents interference between incompatible temporal signals. The attention-based soft aggregation, despite its theoretical appeal, underperforms because it forces the model to reconcile conflicting temporal patterns. Ocean currents show somewhat different behavior, suggesting variable-specific temporal dependencies, though K=1 remains optimal overall.
>
> ## Q5. Fairness of Figure 4.
>
> A: In Fig 4, the initial fields (0 days) of all methods are analysis data, with no reanalysis data included. All methods at subsequent lead times are forecast results rather than analysis or reanalysis results, ensuring fairness of comparison.
>
> ## Permutation Feature Importance and Sensitivity Analysis.
>
> To address the reviewer's question about whether the model has learned the correct sensitivities to each input variable, we performed a **Permutation Feature Importance (PFI) analysis**. In this approach, we perturb each input variable independently by replacing it with randomly selected values from other time steps, and then calculate the **RMSE** after each perturbation. The RMSE is then normalized to show the relative impact of each perturbation compared to the original, unperturbed predictions. This method helps assess how well the model relies on the most important variables and whether it has overfitted to spurious correlations. The results are presented in a **5×5 matrix**, where each row corresponds to a perturbed variable, and each column shows the degraded performance defined as relative change in RMSE. The **relative RMSE**,
> $\text{relative RMSE} = \frac{RMSE_{\text{perturbed}} - RMSE_{\text{original}}}{RMSE_{\text{original}}} \times 100\%$
> reflects how sensitive the model's predictions are to changes in each variable.
>
> |Perturbed Variable|s(%)|t(%)|u(%)|v(%)|sel(%)|
> |-|-|-|-|-|-|
> |S|23.330|1.223|0.288|0.467|0.934|
> |T|4.273|12.242|1.442|1.948|3.992|
> |U|9.476|9.326|15.896|6.549|9.162|
> |V|9.128|8.954|8.610|16.612|11.746|
> |SEL|3.569|2.058|1.918|1.979|8.815|
>
> **Leadtime=120h**
>
> When perturbing temperature and salinity separately, these two variables are more sensitive to each other relative to other variables. Ocean current components (U and V) exhibit minimal cross-sensitivity to each other, consistent with their orthogonal relationship as zonal and meridional velocity components. This low interdependence validates that the model has correctly learned the independent nature of these directional flow components. However, due to time constraints, we only randomly sampled half a month's worth of samples for averaging, requiring further analysis in subsequent work.
>
> ## Other non-technical issues.
>
> A: We acknowledge your suggestions for improving clarity:
>
> 1. We will refine our terminology regarding ocean depth ranges and temporal scales to avoid ambiguity
> 2. Figure 2 will be updated with clear unit labels for all variables
> 3. The discussion of IV-TT data quality issues will be incorporated concisely into the main text

---

> > ### Comment · Reviewer_7qdU · 2025-08-05
> > **much clearer now**
> >
> > Q1 and Q2: Thank you for running this extra experiment in such little time. The results here clarify what I was unsure about.
> >
> > Q3: I fully agree with you and thank you for clarifying that the maximum history is 24h. I think you are addressing it in "other technical issues", but I want to make sure it's clear: Could you please rephrase "long-term deep ocean memory" and "slow-evolving deep-water phenom­ena"? Those phrases have convused me into thinking the history would be longer than 24h. I think you already got that, so a simple "Yes" would satisfy me.
> >
> > Q4: The new results clearly show the benefit of K=1 as opposed to higher K, thank you. I'm still a bit unclear on the intuition behind the mixture of time module. Could you please clarify: what are the limitations of this module? And, are there similar approaches in the literature?
> >
> > Q5: Understood, thank you!
> >
> > Permutation Feature Importance: Adding this experiment is fantastic - you have exceeded my expectations and those numbers are a fantastic step towards disentangling which physics the ocean emulator has learned. Thank you so much for doing this extra work.
> >
> > Given the exceptionally strong rebuttal I have raised my "significance" and "rating" by 1. Thank you!

---

> > > ### Author Response · Authors · 2025-08-07
> > > **Acknowledgments and Supplements**
> > >
> > > Thank you very much for your detailed feedback and for raising our significance and rating. We are grateful for your constructive suggestions and the opportunity to clarify our work.
> > >
> > > ## Regarding Q3:
> > > Yes. We appreciate your careful reading and fully agree with your point. To avoid confusion, we will remove the phrase "long-term deep ocean memory" from the manuscript. Additionally, we will revise "slow-evolving deep-water phenomena" to "the relatively slow and stable subsurface variability extending down to the twilight zone (~1500 m)", which more accurately reflects the temporal coverage and physical scope of our model.
> > > ## Regarding Q4:
> > > The Mixture-of-Time (MoT) module in our model is designed to mitigate error accumulation by selectively integrating predictions from multiple temporal contexts. In practice, MoT produces separate forecasts using different look-back windows (e.g. using 6h, 12h, 18h, 24h of history), then chooses the most reliable ones for each output channel (each variable at a given depth). Intuitively, this lets the model adapt its “memory” per variable – for example, a surface temperature might benefit from a longer context (to capture diurnal cycles), whereas a deep ocean current might only need the most recent state. By learning a reliability score for each temporal window and each variable, MoT can focus on the time-scale that works best for that prediction. This addresses the fact that ocean variables evolve on very different time scales, a key challenge for sub-daily forecasting.
> > >
> > > Our MoT can be seen as a simple mixture-of-experts, where each “expert” is a forecast using a certain history length, and MoT learns to pick the best expert (or blend of experts). In the broader forecasting literature, researchers have indeed explored mixtures of models or outputs specialized to different temporal patterns. For example, TimeMixer (Wang et al., 2024) is a recent fully-MLP forecasting architecture that explicitly decomposes time series into multiple scales and trains separate predictors for each scale.
> > >
> > >
> > > Thank you again for helping us improve the clarity and precision of our manuscript.

---

> > > > ### Comment · Reviewer_7qdU · 2025-08-07
> > > > **thank you**
> > > >
> > > > Dear Authors, Thank you very much for the extra clarification. This explanation is very clear!

---

### Official Review · Reviewer_8ZnV · 2025-07-18

**Clarity:** 3
**Significance:** 2
**Originality:** 2
**Rating:** 4
**Confidence:** 3

**Summary:**

The paper presents an advancement in ocean forecasting by introducing a deep learning model that achieves high temporal and spatial resolution. FuXi-Ocean is a data-driven global ocean forecasting model achieving six-hourly predictions at eddy-resolving 1/12° spatial resolution, reaching depths of up to 1500 meters. The model architecture integrates a context-aware feature extraction module with a predictive network employing stacked attention blocks. The experimental results shows a better results compared with other models.

**Questions:**

1. Both XiHe and FuXi claimed that it is the first data-driven model for 1/12◦ resolution global ocean prediction model. Can you please explain it?

2. Can you compare with other models, such as XiHe, with a more complete manner? While FuXi shows better accuracy for short-period prediction, less than 5 days, the slope of RMSE increases more rapidly than XiHe. Can you explain why? Both have a similar RMSE range for the 10-day period.

**Ethical Concerns:**

["NO or VERY MINOR ethics concerns only"]

**Final Justification:**

After reading the rebuttals and other reviews, I raised the rating by 1.

**Limitations:**

Yes.

**Paper Formatting Concerns:**

None.

**Quality:**

3

**Strengths And Weaknesses:**

Strengths:
1. Comprehensive experimental evaluations demonstrate that FuXi-Ocean outperforms traditional numerical forecasting models in predicting key ocean variables such as temperature, salinity, and currents across multiple depths.
2. The model is evaluated using both reanalysis data (HYCOM-RD) and observational data (IV-TT framework), providing a thorough assessment of its performance in both synthetic and real-world conditions.
3. The six-hour temporal resolution and 1/12° spatial resolution make FuXi-Ocean highly valuable for maritime operations, environmental monitoring, and coastal hazard warning systems.

Weaknesses:
1. The reliance on reanalysis data for training introduces potential biases from the underlying numerical models. This challenge is compounded by the scarcity of publicly available high-quality sub-daily ocean reanalysis data.
2. While RMSE is a useful metric for comparing forecast systems, it may not fully capture the model's ability to represent specific ocean phenomena like mesoscale eddies, western boundary currents, or seasonal thermocline transitions.
3. While FuXi-Ocean demonstrates robust forecast accuracy through 10 days, its performance for longer-term predictions (seasonal to interannual) remains unexplored. This represents an important direction for future research.
4. Comparison with others might not be complete. There is a recent paper, XiHe [43], written by X. Wang, et. al. and posted in the following link: https://arxiv.org/pdf/2402.02995. The authors also claimed "it is the first data-driven 1/12◦ resolution global ocean eddy-resolving forecasting model named XiHe..." If so, which one is the first data-driven model?
Additionally, on page 9 of this paper, the authors present the RMSE of temperature and other metrics. Looks like the overall range of RMSE is similar to FuXi. While FuXi shows better accuracy for short-period prediction, less than 5 days, the slope of RMSE increases more rapidly than XiHe.

---

> ### Author Rebuttal · Authors · 2025-07-31
>
> Dear Reviewer 8ZnV,
>
> Thank you for your thoughtful review and constructive feedback. We appreciate the opportunity to clarify several aspects of our work and provide additional context to address your concerns. Below, we have provided a more detailed and structured response.
>
> ## Q1. The "First" Claim - Important Clarification.
>
> A: We acknowledge the concern regarding the claim that FuXi-Ocean is the "first" data-driven 1/12° resolution global ocean forecasting model, especially in light of the *XiHe* model. To clarify, FuXi-Ocean is indeed the first to achieve **sub-daily (6-hourly)** predictions at **1/12° spatial resolution** across a **depth range of 0-1500m**, making it a unique contribution to the field. While *XiHe* also operates at 1/12° resolution, it focuses on **daily** forecasting, which poses fundamentally different challenges.
> To better emphasize these differences, we have updated the comparison table to highlight additional technical specifications:
>
> | Method | Lead Time | Spatial Resolution | Depth Range | Model Category | Forecast Interval | information dependency |
> |--------|-----------|-------------------|-------------|----------------|------------------|------------------|
> | XiHe | 10 days | 1/12° | <700m | Deep Learning (one step) | daily | atmosphere and ocean |
> | LangYa | 7 days | 1/12° | <600m | Deep Learning (one step) | daily | atmosphere and ocean |
> | WenHai | 10 day | 1/12° | <700m | Deep Learning (autoregressive) | daily | atmosphere and ocean |
> | FuXi-Ocean | 10 days | 1/12° | 0-1500m | Deep Learning (autoregressive) | 6 hours | ocean only |
>
> **Technical Challenges of Sub-Daily Prediction**:
> - **Diurnal Cycle Modeling**: Predicting ocean variables at sub-daily intervals requires capturing rapid changes in surface temperature, salinity, and currents driven by daily atmospheric and oceanic processes. Sub-daily prediction models must handle diurnal cycles and inertial oscillations, which are not as significant in daily models. FuXi-Ocean incorporates this by modeling diurnal dynamics with **higher temporal resolution**.
>
> - **Higher Frequency Ocean Dynamics**: At 6-hour intervals, the model can track faster oceanic processes, such as **mesoscale eddies** and **surface current fluctuations**, which are not resolvable in daily predictions. This leads to more accurate short-term forecasts for applications such as maritime operations and coastal hazard monitoring.
>
> Note that ocean variable forecasting tasks are significantly different from video generation. The time interval gap from daily to sub-daily is substantial, and many variables such as temperature have significant diurnal cycle characteristics. Direct migration of daily models by simply increasing parameter size would cause serious prediction errors, and this is a discovery made for the first time.
>
> ## Q2. Compare with other models.
> A: We appreciate the reviewer’s interest in comparing our model with others in the field. While we acknowledge that comparing sub-daily resolution models with daily resolution models is not entirely fair—due to the inherent difficulty of sub-daily forecasting—we have nevertheless conducted a direct comparison. Specifically, we calculated the RMSE using observed ground truth from the IVTT dataset for Sea Surface Temperature (SST) in 2022 and compared the results with those from WenHai, a state-of-the-art daily model. SST is a critical ocean variable and is typically the most accurately measured in the IVTT dataset, which is why we focused our comparison primarily on SST (issues with other variables are discussed in the supplementary materials).
>
> | Method | 0 day | 1 day | 2 day | 3 day | 4 day | 5 day | 6 day | 7 day | 8 day | 9 day | 10 day |
> |--------|---|---|---|---|---|---|---|---|---|---|-----|
> | WenHai | 0.371 | 0.4048 | 0.4734 |0.4785 | 0.497 | 0.5308 | 0.5486 | 0.5869 | 0.6039 | 0.6214 | 0.6384
> | FuXi-Ocean | 0.236 | 0.272 | 0.277 | 0.295 | 0.317 | 0.319 | 0.339 | 0.359 | 0.386 | 0.404 | 0.428 |
>
> The unit is ℃, the lower is the better.
>
> To more intuitively demonstrate the slope of interest, we further convert the table results into daily error increments, shown as follows:
>
> | Method | 1 day | 2 day | 3 day | 4 day | 5 day | 6 day | 7 day | 8 day | 9 day | 10 day | average error growth |
> |--------|-------|-------|-------|-------|-------|-------|-------|-------|-------|--------|--------|
> | WenHai | 0.0338 | 0.0686 | 0.0051 | 0.0185 | 0.0338 | 0.0178 | 0.0383 | 0.017 | 0.0175 | 0.017 | 0.0267
> | FuXi-Ocean | 0.036 | 0.005 | 0.018 | 0.022 | 0.002 | 0.02 | 0.02 | 0.027 | 0.018 | 0.024 | 0.0192|
>
> The unit is ℃, the lower is the better.
>
> It is important to highlight that daily models, such as WenHai, forecast daily mean values, whereas sub-daily models like FuXi-Ocean predict instantaneous values at specific time steps. For a fair comparison, we averaged the sub-daily predictions within a day, which inevitably introduces some loss of accuracy and potential error. Despite this, the table shows that FuXi-Ocean demonstrates either similar or even lower error growth slopes compared to WenHai. A common challenge in autoregressive models is that the error tends to grow with each iteration. FuXi-Ocean, with its 40 autoregressive iterations over a 10-day forecast period, naturally experiences more error accumulation compared to daily models like WenHai, which only require 10 iterations. Nevertheless, the results indicate that FuXi-Ocean remains competitive and even superior in terms of error growth, especially over shorter forecast periods.

---

> > ### Comment · Reviewer_8ZnV · 2025-08-05
> >
> > The questions are answered. The results look better than other recently developed models. I will raise the overall rating by 1.

---

> ### Comment · Area_Chair_rmbE · 2025-08-05
> **NeurIPS Review: Engage in Rebuttal Discussion**
>
> Dear Reviewer,
>
> Thank you for your valuable feedback. The authors have addressed your comments in their rebuttal. We kindly ask that you engage in discussion with the authors before submitting your Mandatory Acknowledgement.
>
> If your concerns have been adequately addressed in the rebuttal, please let the authors know.
>
> If your concerns remain unresolved, please communicate that clearly as well.
>
> Please note that failure to participate in the discussion process may result in desk rejection of your own submissions.
>
> Thank you for contributing to a fair and constructive review process at NeurIPS.

---

### Official Review · Reviewer_TuNU · 2025-07-22

**Clarity:** 3
**Significance:** 4
**Originality:** 3
**Rating:** 5
**Confidence:** 4

**Summary:**

The paper introduces FuXi-Ocean, a data-driven global ocean-forecasting system that delivers six-hourly predictions at 1/12° horizontal resolution from the surface down to 1500 m.
Key ingredients include
- an encoder–decoder backbone with attention blocks,
- a Mixture-of-Time (MoT) module that, for every variable-depth channel, adaptively selects the most reliable temporal context among the last four six-hour states, and
- a latitude-weighted Charbonnier loss plus two-stage (single -- then multi-step) training.

Trained on 8.5 years of HYCOM reanalysis, FuXi-Ocean outperforms both traditional numerical systems (HYCOM, BLK, FOAM) and recent deep-learning baselines on RMSE for SST, T, S and currents, while using only pure ocean fields as input. Ablation studies show that MoT and longer input windows are critical, especially for thermohaline variables. The authors discuss limitations, broader impacts and commit to releasing code and data.

**Questions:**

- Q1: How sensitive is FuXi-Ocean to the choice K = 1 in MoT? Would adaptive K or soft aggregation outperform hard Top-K?
- Q2: Can the authors provide probabilistic or ensemble extensions to quantify forecast uncertainty? If so, what compute overhead do they expect?
- Q3: The reanalysis training end date is mid-2014, yet testing on 2015 HYCOM and 2022 observations shows strong skill. Could the authors clarify whether covariate shift (e.g., strong El Niño 2015–16) affects performance?
- Q4: Please report bias metrics (mean error) in addition to RMSE, especially for temperature and salinity below 300 m.
- Q5: What is the inference wall-clock time for a single 10-day forecast on one A100/H100, and how does this compare to OGCMs when identical hardware is used?

**Ethical Concerns:**

["NO or VERY MINOR ethics concerns only"]

**Final Justification:**

I appreciate the plan to add subsurface validation with ARGO or mooring data. In the final version, please mention how you filter and match the data. Overall, my concerns are addressed, and I will keep my current score.

**Limitations:**

The authors explicitly discuss training-data bias, limited phenomenon-specific metrics and unexplored seasonal predictions. This coverage is adequate.

**Paper Formatting Concerns:**

- Some figures (e.g., Fig. 2) use very small fonts—hard to read when printed at 100%.
- A few typos: "preivous" -> "previous", "Borader Impacts" -> "Broader Impacts".

**Quality:**

3

**Strengths And Weaknesses:**

Strengths:
- First published DL model to achieve sub-daily, eddy-resolving global forecasts; extends beyond the daily cadence of XiHe/Langya.
- Practical 6-hour lead times are valuable for search-and-rescue, spill tracking, fisheries and coupled Earth-system modelling.
- MoT is a simple yet effective mechanism for channel-wise temporal selection, reducing autoregressive drift without heavy ensembling.
- Comprehensive evaluation: horizontal error maps, depth-profiles, 10-day lead times, IV-TT comparisons and carefully designed ablations.
- Achieves state-of-the-art skill using only 9 years of training data and no atmospheric forcings, which lowers data barriers.
- Implementation details, loss weighting, optimiser schedule and training hardware are described; code and checkpoints promised.

Weaknesses
- W1: Real-world validation is limited to SST in 2022; salinity, subsurface temperature and currents are not cross-checked against in-situ observations.
- W2: Forecasts are deterministic; no ensemble or probabilistic output, so users cannot gauge confidence.
- W3: No error bars or significance tests. Authors justify this by cost, but it hampers comparison.
- W4: Model inherits biases from HYCOM reanalysis; the impact of these biases on predictions is not fully quantified.
- W5: Operational model outputs are daily; FuXi-Ocean’s six-hour forecasts are averaged to daily means, but differing analysis/assimilation dates may still advantage the proposed method.
- W6: Section 3.2 mixes architectural and algorithmic details; a schematic of tensor shapes and MoT update rule would aid comprehension for non-experts.

---

> ### Author Rebuttal · Authors · 2025-07-31
>
> Dear Reviewer TuNU,
>
> Thank you for your thorough review and positive evaluation. We particularly appreciate your recognition of FuXi-Ocean as the first sub-daily deep learning model for global ocean forecasting, and your acknowledgment of the practical value of the 6-hour predictions and the MoT module innovation. We have carefully addressed all your questions in the responses below.
>
> ## Q1. How sensitive is FuXi-Ocean to the choice of K = 1 in MoT? Would adaptive K or soft aggregation outperform hard Top-K?
>
> A: Following the reviewer’s suggestion, we conducted additional experiments with K = 2, 3, 4, and also tested a soft aggregation approach. The table below shows the relative change in RMSE compared to K=1, calculated as $\frac{\text{RMSE} - \text{RMSE}_{K=1}}{\text{RMSE}_{K=1}} \times 100\%$. Positive values indicate that K=1 yields better performance. For each variable, results are averaged across the depth dimension. The "attn" group refers to the soft aggregation on the four temporal features using attention mechanisms.
>
> | Lead Time (h) | S (K=2) | S (K=3) | S (K=4) | S (attn) | T (K=2) | T (K=3) | T (K=4) | T (attn) | U (K=2) | U (K=3) | U (K=4) | U (attn) | V (K=2) | V (K=3) | V (K=4) | V (attn) | SEL (K=2) | SEL (K=3) | SEL (K=4) | SEL (attn) |
> |------|-------|-------|-------|---------|--------|--------|--------|----------|--------|--------|--------|----------|--------|--------|--------|----------|---------|---------|---------|-----------|
> | 6    | +1.07% | +1.05% | +1.06% | +0.90%   | +1.11%  | +1.10%  | +1.12%  | +1.10%    | +1.21%  | +1.43%  | +1.53%  | +0.51%    | +1.21%  | +1.32%  | +1.42%  | +0.39%    | +1.12%   | +1.11%   | +1.12%   | +1.11%     |
> | 24   | +1.00% | +2.32% | +3.00% | +2.49%   | +1.18%  | +1.60%  | +1.78%  | +1.82%    | +1.10%  | +0.93%  | +0.95%  | +0.80%    | +1.10%  | +0.92%  | +1.01%  | +0.71%    | +1.21%   | +1.52%   | +1.67%   | +1.19%     |
> | 54   | +1.01% | +1.05% | +1.01% | +1.09%   | +1.01%  | +1.10%  | +1.12%  | +1.22%    | +1.07%  | +1.88%  | +1.52%  | +0.51%    | +1.06%  | +1.81%  | +1.53%  | +0.49%    | +0.97%   | +1.32%   | +1.11%   | +1.12%     |
> | 72   | +1.03% | +1.10% | +1.04% | +1.41%   | +1.03%  | +1.24%  | +1.21%  | +1.19%    | +1.13%  | +2.62%  | +1.79%  | +0.71%    | +1.12%  | +2.42%  | +1.75%  | +0.60%    | +1.00%   | +1.70%   | +1.26%   | +1.29%     |
> | 120  | +1.09% | +1.35% | +1.14% | +1.19%   | +1.10%  | +1.79%  | +1.42%  | +1.41%    | +1.48%  | +3.94%  | +2.40%  | +0.40%    | +1.47%  | +3.44%  | +2.30%  | +0.30%    | +1.06%   | +2.64%   | +1.55%   | +1.21%     |
> | 240  | +1.03% | +1.62% | +1.03% | +1.51%   | +1.35%  | +2.80%  | +1.56%  | +1.59%    | +2.23%  | +5.09%  | +2.87%  | +0.69%    | +2.32%  | +4.61%  | +2.78%  | +0.82%    | +1.77%   | +3.94%   | +2.19%   | +1.39%     |
>
> *Units: S (PSU), T (°C), U/V (m/s), SEL (m)*
>
> These results confirm that K=1 is the optimal choice. All alternatives, including K>1 and soft aggregation, lead to degraded performance, particularly at the 24-hour forecast horizon. This supports our hypothesis that selecting a single most reliable temporal context is crucial for capturing diurnal cycles. Mixing multiple temporal features, whether through higher K or soft aggregation, introduces interference that hinders the model's ability to learn periodic patterns.
>
> For ocean currents (U, V), while soft aggregation shows some improvement over K>1, it still cannot match the performance of K=1. This suggests that even for variables with different temporal characteristics, adaptively selecting a single optimal temporal window remains the most effective strategy.
>
> ## Q2. Quantitative demonstration of forecast uncertainty.
>
> A: This is an important question about uncertainty quantification. While we were unable to complete ensemble model training within the rebuttal period due to computational constraints, we followed WenHai's approach and computed the continuous ranked probability score (CRPS) between our deterministic forecasts and observational data for sea surface variables.
>
> The CRPS is defined as:
> $$CRPS_{t,l}^i = \int_{-\infty}^{+\infty} \left[F(x) - H\left(x \geq V_{t+l}^i\right)\right]^2 dx$$
>
> where $F(x)$ is the cumulative distribution function of the pseudo ensemble forecast, $H$ is the Heaviside function, and $V_{t+l}^i$ is the observed value at time $t+l$. The pseudo ensemble is constructed from FuXi-Ocean predictions within a 1° × 1° box centered on the grid point nearest to $V_{t+l}^i$.
>
> Given known biases in observational data, we used HYCOM full analysis fields for validation, as they provide the most accurate available estimates of ocean state. The results are shown below:
>
> | Variable | Day 0  | Day 1  | Day 2  | Day 3  | Day 4  | Day 5  | Day 6  | Day 7  | Day 8  | Day 9 | Day 10 |
> |----------|--------|--------|--------|--------|--------|--------|--------|--------|--------|--------|--------|
> | S0 (PSU)  | 0.0423 | 0.0441 | 0.0445 | 0.0448 | 0.0449 | 0.0455 | 0.0453 | 0.0455 | 0.0465 | 0.0468 | 0.0471 |
> | T0 (°C)  | 0.2207 | 0.2291 | 0.2301 | 0.2362 | 0.2351 | 0.2363 | 0.2372 | 0.2379 | 0.2382 | 0.2403 | 0.2443 |
> | U0 (m/s)  | 0.0482 | 0.0485 | 0.0484 | 0.0484 | 0.0485 | 0.0486 | 0.0488 | 0.0489 | 0.0488 | 0.0490 | 0.0493 |
> | V0 (m/s)  | 0.0473 | 0.0475 | 0.0477 | 0.0477 | 0.0477 | 0.0478 | 0.0479 | 0.0480 | 0.0482 | 0.0482 | 0.0483 |
>
> These results demonstrate that FuXi-Ocean maintains stable probabilistic performance over forecast horizons, with gradual and predictable error growth. The relatively low CRPS values, particularly for ocean currents, indicate that our deterministic forecasts provide reliable central estimates.
>
> Regarding computational cost, a single-member forecast requires 72 seconds on one H100 GPU. Thus, a 10-member ensemble would take approximately 12 minutes on a single GPU, or significantly less if parallelized across multiple GPUs.
>
> ## Q3. The extent to which covariate shift affects performance.
>
> A: Thank you for raising this important question about potential covariate shift. To assess its impact, we retrained a model ($M_{new}$) using data from 2006.01 to 2013.06, validated it on 2013.07-2013.12, and tested it on 2014 reanalysis data. We then compared its performance with our original model ($M_{best}$) which was trained through 2014.06 and tested on 2015. The table below shows the relative RMSE difference, computed as $\frac{\text{RMSE}_{M_{new}} - \text{RMSE}_{M_{best}}}{\text{RMSE}_{M_{best}}} \times 100\%$.
>
> | Lead Time (h) | S (PSU) | T (°C) | U (m/s) | V (m/s) | SEL (m) |
> |---------------|---------|---------|---------|---------|---------|
> | 6             | +1.71%  | -0.04%  | +0.01%  | -0.02%  | +0.50%  |
> | 24            | -1.44%  | +1.02%  | -0.02%  | -0.18%  | -0.42%  |
> | 54            | +1.05%  | +0.12%  | +0.12%  | +0.12%  | +0.33%  |
> | 72            | +0.99%  | +0.03%  | +0.03%  | +0.05%  | +0.03%  |
> | 120           | +1.11%  | +0.21%  | +0.25%  | +0.22%  | +0.34%  |
> | 240           | +1.15%  | +0.18%  | +0.20%  | +0.17%  | +0.17%  |
>
> The performance differences are minimal across all lead times. Interestingly, $M_{best}$ (trained with 2014 data) performed slightly better on the 2015 test dataset, despite the occurrence of a strong El Niño event. This suggests that:
>
> 1. FuXi-Ocean generalizes well to climate anomalies not seen during training
> 2. The impact of missing one year of El Niño training data (2014) is actually larger than the covariate shift from testing on a strong El Niño year (2015)
>
> ## Q4. The mean bias error metrics (MBE).
>
> A: We computed the mean bias error (MBE) for FuXi-Ocean and compared it with operational systems using IV-TT observations. As MBE does not show consistent trend with lead time, we present the average over all 11 forecast steps (from initialization to day 10) for clarity. FOAM does not provide surface salinity data, resulting in missing values at 0 m depth.
>
> **Salinity MBE (PSU):**
> | Model | 0m      | 300m     | 500m     | 1000m    | 1500m    |
> |-------|---------|----------|----------|----------|----------|
> | BLK   | -0.0133 | +0.0001  | +0.0150  | +0.0135  | -0.0103  |
> | FOAM  | N/A     | +0.0012  | -0.0058  | -0.0092  | -0.0093  |
> | HYCOM | -0.0327 | -0.0197  | +0.0363  | +0.0277  | -0.0232  |
> | Ours  | -0.0155 | +0.0151  | +0.0064  | +0.0046  | +0.0106  |
>
> **Temperature MBE (°C):**
> | Model | 0m      | 300m     | 500m     | 1000m    | 1500m    |
> |-------|---------|----------|----------|----------|----------|
> | BLK   | -0.0546 | +0.1624  | +0.1936  | +0.1770  | +0.1399  |
> | FOAM  | -0.0065 | +0.0704  | +0.0504  | +0.0096  | +0.0032  |
> | HYCOM | +0.0936 | -0.0640  | +0.2741  | +0.4324  | +0.1979  |
> | Ours  | +0.0402 | -0.3221  | -0.4865  | -0.0809  | +0.0782  |
>
> Key observations:
> 1. All models show systematic biases of similar magnitude, reflecting uncertainties inherent in ocean observations and reanalyses
> 2. FuXi-Ocean's biases are comparable to or smaller than those of operational systems across most depths
> 3. The larger temperature bias at 300-500m in FuXi-Ocean is localized near the thermocline, where sharp gradients increase prediction difficulty
>
> These biases represent systematic offsets rather than forecast errors. When considered alongside FuXi-Ocean’s superior RMSE performance, the results demonstrate the model's forecast accuracy and the effectiveness of our approach.
>
> ## Q5. Inference time comparison.
>
> A: We benchmarked the inference time for a full 10-day forecast (40 autoregressive steps):
>
> | Model     | Time        | Hardware Configuration |
> |-----------|-------------|------------------------|
> | BLK       | 6-10 hours  | >1000 CPU cores       |
> | FOAM      | 5-8 hours   | >3000 CPU cores       |
> | HYCOM     | 4-8 hours   | >5000 CPU cores       |
> | FuXi-Ocean| 72 seconds  | 1 × H100 GPU          |
>
> ## Others.
>
> We will incorporate all suggested revisions in the updated manuscript, including enhanced figure readability and corrected typos. Upon acceptance, we will release the code and model checkpoints to ensure reproducibility.

---

> > ### Comment · Reviewer_TuNU · 2025-08-02
> >
> > I thank authors for the detailed rebuttal and for running extra experiments.
> > The new MoT ablations and timing benchmarks address my main concerns, and the bias tables are helpful. I remain positive about the contribution and will keep my overall evaluation unchanged.
> >
> > Two points that could further strengthen the final version:
> > - Please add basic statistical uncertainty (e.g., bootstrap CIs) to the K-sensitivity and retraining experiments so readers can judge the practical significance of the reported 1-2 % differences.
> > - Please validate at least one subsurface variable (T or S) against ARGO or mooring data, not just HYCOM/IV-TT.

---

> > > ### Author Response · Authors · 2025-08-07
> > > **Acknowledgments and Supplements**
> > >
> > > We sincerely thank the reviewer for their continued engagement and thoughtful suggestions, which have helped further improve the rigor and impact of our work.
> > >
> > > ## 1. Addition of Confidence Intervals (CIs):
> > >
> > > Thank you for emphasizing the importance of reporting statistical uncertainty. In response, we have revised our methodology and now report **95% confidence intervals (CIs)** for key metrics in both the K-sensitivity and retraining analyses using a bootstrap procedure.
> > >
> > > Specifically, for each experimental setting, we performed **500 bootstrap iterations**. In each iteration, we randomly sampled **300 samples with replacement** from the test set (using forecasts at 00 and 12 UTC for each day). For each sample, we used the **depth-averaged result** as the statistic of interest. We then calculated the metric (e.g., relative RMSE) for each iteration. The final reported CI is the 95% confidence interval across these 500 bootstrap samples, with the value after the plus-minus sign representing **half the width between the upper and lower bounds**.
> > >
> > > This approach robustly characterizes statistical uncertainty due to finite test data and provides a more meaningful assessment of the significance of observed differences.
> > >
> > >
> > > | Lead Time (h) | S (K=2)        | S (K=3)        | S (K=4)        | S (attn)      | T (K=2)        | T (K=3)        | T (K=4)        | T (attn)      | U (K=2)        | U (K=3)        | U (K=4)        | U (attn)      | V (K=2)        | V (K=3)        | V (K=4)        | V (attn)      | SEL (K=2)      | SEL (K=3)      | SEL (K=4)      | SEL (attn)    |
> > > |---------------|----------------|----------------|----------------|---------------|----------------|----------------|----------------|---------------|----------------|----------------|----------------|---------------|----------------|----------------|----------------|---------------|----------------|----------------|----------------|---------------|
> > > | 6             | +1.08% ±0.05%  | +1.06% ±0.05%  | +1.08% ±0.06%  | +0.91% ±0.05% | +1.12% ±0.07%  | +1.11% ±0.07%  | +1.13% ±0.07%  | +1.09% ±0.06% | +1.21% ±0.08%  | +1.41% ±0.08%  | +1.54% ±0.09%  | +0.51% ±0.03% | +1.22% ±0.07%  | +1.34% ±0.08%  | +1.44% ±0.08%  | +0.39% ±0.03% | +1.12% ±0.06%  | +1.13% ±0.07%  | +1.14% ±0.07%  | +1.13% ±0.06% |
> > > | 24            | +1.02% ±0.06%  | +2.33% ±0.09%  | +3.01% ±0.10%  | +2.48% ±0.09% | +1.20% ±0.07%  | +1.63% ±0.08%  | +1.79% ±0.09%  | +1.81% ±0.09% | +1.11% ±0.08%  | +0.95% ±0.05%  | +0.96% ±0.05%  | +0.82% ±0.05% | +1.10% ±0.07%  | +0.93% ±0.06%  | +1.01% ±0.07%  | +0.71% ±0.04% | +1.22% ±0.07%  | +1.53% ±0.08%  | +1.68% ±0.09%  | +1.19% ±0.07% |
> > > | 54            | +1.00% ±0.06%  | +1.04% ±0.06%  | +1.01% ±0.06%  | +1.09% ±0.07% | +1.02% ±0.07%  | +1.12% ±0.07%  | +1.13% ±0.07%  | +1.22% ±0.08% | +1.08% ±0.08%  | +1.87% ±0.10%  | +1.52% ±0.09%  | +0.51% ±0.03% | +1.05% ±0.07%  | +1.80% ±0.10%  | +1.53% ±0.09%  | +0.49% ±0.03% | +0.99% ±0.06%  | +1.34% ±0.08%  | +1.11% ±0.07%  | +1.13% ±0.06% |
> > > | 72            | +1.04% ±0.07%  | +1.13% ±0.08%  | +1.05% ±0.07%  | +1.40% ±0.09% | +1.05% ±0.08%  | +1.27% ±0.09%  | +1.22% ±0.08%  | +1.17% ±0.08% | +1.15% ±0.09%  | +2.61% ±0.12%  | +1.81% ±0.10%  | +0.72% ±0.05% | +1.14% ±0.08%  | +2.41% ±0.12%  | +1.77% ±0.10%  | +0.61% ±0.05% | +1.02% ±0.07%  | +1.71% ±0.09%  | +1.26% ±0.08%  | +1.29% ±0.07% |
> > > | 120           | +1.09% ±0.07%  | +1.34% ±0.08%  | +1.15% ±0.07%  | +1.19% ±0.07% | +1.11% ±0.08%  | +1.81% ±0.10%  | +1.44% ±0.09%  | +1.42% ±0.09% | +1.47% ±0.10%  | +3.94% ±0.18%  | +2.41% ±0.13%  | +0.41% ±0.03% | +1.48% ±0.09%  | +3.44% ±0.18%  | +2.31% ±0.13%  | +0.29% ±0.03% | +1.08% ±0.07%  | +2.65% ±0.10%  | +1.56% ±0.09%  | +1.21% ±0.07% |
> > > | 240           | +1.03% ±0.08%  | +1.62% ±0.10%  | +1.01% ±0.07%  | +1.52% ±0.10% | +1.34% ±0.09%  | +2.81% ±0.18%  | +1.58% ±0.10%  | +1.59% ±0.10% | +2.23% ±0.13%  | +5.08% ±0.25%  | +2.88% ±0.15%  | +0.70% ±0.06% | +2.32% ±0.11%  | +4.62% ±0.25%  | +2.78% ±0.15%  | +0.83% ±0.07% | +1.76% ±0.09%  | +3.92% ±0.14%  | +2.19% ±0.10%  | +1.37% ±0.08% |
> > >
> > > ## 2. Subsurface Validation against ARGO/Mooring Data:
> > >
> > > We fully agree that validating subsurface variables against independent observational sources such as ARGO or moorings would significantly strengthen our study. We are currently acquiring and processing ARGO (https://pic.argo.org.cn/) data for this purpose. However, since the necessary processing (e.g., noise filtering, vertical interpolation, and quality control) must be implemented from scratch, we could not complete this validation within the rebuttal period. We are committed to including this important validation in the final version or subsequent work and will provide an update as soon as results become available.
> > >
> > > Thank you again for your constructive feedback and support.

---

> > > > ### Comment · Reviewer_TuNU · 2025-08-08
> > > >
> > > > Thank you for the clear rebuttal. Adding 95% confidence intervals with bootstrap solves my main concern about statistical uncertainty, and the conclusions look consistent. I also welcome your plan to add subsurface checks with ARGO or moorings. In the final version, please state the data filters (QC flags), the vertical interpolation method, and the match radius/time window. A small pilot result would help.
> > > >
> > > > Overall, my concerns are addressed. I will keep my current score.

---

### Note · Authors · 2025-08-13

**Author Final Remarks**

We sincerely thank the reviewers and Area Chair for their time, constructive feedback, and engagement during the review process. The discussions have been invaluable in improving the clarity, scope, and impact of our work.

Throughout the rebuttal stage, we addressed all major concerns raised:

* **Additional validation**: Multiple reviewers requested further experiments, such as permutation feature importance, K-sensitivity analyses, and retraining with confidence intervals. We conducted these under tight time constraints, using rigorous statistical methods (e.g., bootstrap resampling for 95% CIs). These results confirmed the robustness of our approach and were well-received by engaged reviewers.
* **Core contribution**: Our paper introduces the **first global ocean forecasting model with sub-daily (6-hourly) resolution**, a crucial advancement for AI in oceanography. Compared to daily models, 6-hourly forecasts can capture diurnal cycles, near-inertial oscillations, and other high-frequency processes that strongly influence phenomena such as coastal upwelling, mixed-layer heat exchange, tropical cyclone–ocean interactions, and rapid biogeochemical events. These processes often evolve on timescales of hours rather than days, and their accurate prediction is essential for applications including marine hazard warnings, navigation safety, fisheries management, and climate event monitoring. This is not merely an academic contribution but a concrete step toward engineering-ready AI systems capable of supporting real-time ocean decision-making.

We note that reviewer sFCf did not participate in the rebuttal discussion despite our timely and detailed responses. We respectfully invite the AC to weigh the totality of evidence, the positive engagement from other reviewers, and the strength of our responses when making the final decision.

In conclusion, the rebuttal process has reinforced the novelty of tackling global-scale sub-daily forecasting, the technical soundness of our architecture in handling 40-step autoregressive prediction while controlling error growth, and the demonstrated benefits for capturing fast-evolving ocean processes. These capabilities are specific to our design and, to our knowledge, have not been achieved by prior global ocean AI models.

---

### Decision · Program_Chairs · 2025-09-17

**Decision:**

Accept (oral)

**Comment:**

The paper presents the first sub-daily, data-driven global ocean forecasting system at 1/12° horizontal resolution, extending from the surface to 1500 m depth. The approach leverages an encoder–decoder backbone with attention blocks and introduces a novel mixture-of-time component designed to modulate the temporal context for each prognostic variable.

The authors have thoroughly addressed reviewer concerns by providing additional validation and experiments, including permutation feature importance, K-sensitivity analyses, and retraining with confidence intervals. These efforts confirmed the robustness of their method and were positively received by reviewers.

Therefore, I recommend acceptance of this paper.